# Recent Advances in Age-Related Macular Degeneration Therapies

**DOI:** 10.3390/molecules27165089

**Published:** 2022-08-10

**Authors:** Marie Fabre, Lou Mateo, Diana Lamaa, Stéphanie Baillif, Gilles Pagès, Luc Demange, Cyril Ronco, Rachid Benhida

**Affiliations:** 1Institut de Chimie de Nice UMR 7272, Université Côte d’Azur, CNRS, 06108 Nice, France; 2CiTCoM, UMR 8038 CNRS, Faculté de Pharmacie, Université de Paris Cité, 4, Avenue de l’Observatoire, 75006 Paris, France; 3Ophthalmology Department, University Hospital of Nice, 30 Avenue De La Voie Romaine, 06000 Nice, France; 4Institute for Research on Cancer and Aging (IRCAN), UMR 7284 and INSERM U 1081, Université Côte d’Azur, CNRS 28 Avenue de Valombrose, 06107 Nice, France; 5Department of Chemical and Biochemical Sciences-Green Process Engineering (CBS-GPE), Mohamed VI Polytechnic University (UM6P), Benguerir 43150, Morocco

**Keywords:** age-related maculopathy, dry age-related macular degeneration, wet age-related macular degeneration, eye’s disease, elderly, clinical trials

## Abstract

Age-related macular degeneration (AMD) was described for the first time in the 1840s and is currently the leading cause of blindness for patients over 65 years in Western Countries. This disease impacts the eye’s posterior segment and damages the macula, a retina section with high levels of photoreceptor cells and responsible for the central vision. Advanced AMD stages are divided into the atrophic (dry) form and the exudative (wet) form. Atrophic AMD consists in the progressive atrophy of the retinal pigment epithelium (RPE) and the outer retinal layers, while the exudative form results in the anarchic invasion by choroidal neo-vessels of RPE and the retina. This invasion is responsible for fluid accumulation in the intra/sub-retinal spaces and for a progressive dysfunction of the photoreceptor cells. To date, the few existing anti-AMD therapies may only delay or suspend its progression, without providing cure to patients. However, in the last decade, an outstanding number of research programs targeting its different aspects have been initiated by academics and industrials. This review aims to bring together the most recent advances and insights into the mechanisms underlying AMD pathogenicity and disease evolution, and to highlight the current hypotheses towards the development of new treatments, i.e., symptomatic vs. curative. The therapeutic options and drugs proposed to tackle these mechanisms are analyzed and critically compared. A particular emphasis has been given to the therapeutic agents currently tested in clinical trials, whose results have been carefully collected and discussed whenever possible.

## 1. Introduction

In the last century, the increase in life expectancy led to the emergence of new public health problems related to the high prevalence of age-related pathologies. Among them, age-related macular degeneration (AMD) is a degenerative disease associated with the aging of the macula [1]. Nowadays, AMD affects more than 50% of people over 80 worldwide. Currently, no curative treatments for AMD exist, the only proposed cares are limited to slowing down its progression. AMD is one of the leading causes of blindness in developed countries. Therefore, the search for new AMD treatments has grown considerably during the last decade. However, their development remains very challenging because the pathogenesis and the progression of the disease to advanced stages have not been completely elucidated. In consequence, several clinical trials have been launched, first arousing hopes, but most of them have been stopped due to poor clinical outcomes. This review aims at providing to the reader a comprehensive and critical overview of relevant investigations, mechanisms, and hypotheses. The main therapeutic agents investigated by the scientific community are depicted, in order to point out the most promising direction toward the development of efficient and safe treatments in a near future. 

The eye is the central organ of the visual system, responsible of the phototransduction mechanism. It consists of the switch of captured light into a cellular signal, and its transmission to the brain through the optic nerve [2,3].

The eyeball is divided into two segments that encompass different compartments and envelopes with distinct and specific functions: (i) the anterior segment includes the ciliary body, the cornea, the iris, and the lens, while (ii) the posterior segment includes the choroid, the retina and the optic nerve. In the center of the retina, more precisely in the macula, is located the highest concentration of photoreceptors, responsible for both color perception and vision clarity. Macula therefore allow maximal visual acuity (VA) [4]. In addition, several envelopes protect the eye globe, and maintain the eye’s shape. Among them, we could distinguish the sclera, a white resistant fibrous tissue that protects the eye, as well as the conjunctiva, a very thin membrane covering the sclera at its front (Figure 1) [3]. 

Worldwide, according to World Health Organization (WHO), approximately 2.2 billion people suffer a form of vision impairment caused by different eye pathologies [5]. We can distinguish two main categories of pathologies: the general diseases affecting the eye in its globality, and the specific pathologies restricted to one compartment or membrane. Three diseases, cataracts (20%), glaucoma (20.5%), and AMD (26%), are responsible for more than two-thirds of patients suffering from eye pathologies (Figure 2) [6]. 

Cataracts affects the crystalline lens resulting in lens opacity, that usually affects both eyes [6]. The only effective treatment consists in the surgical replacement of the crystalline lens by an artificial one. On the other hand, glaucoma is a medical emergency, caused by an abnormal accumulation of fluids in the eye’s anterior fragment, leading to an increased intraocular pressure in its posterior segment and to a degeneration and necrosis of the optic nerve cells. The glaucoma’s causes remain not fully understood, and treatments are rare, sometime only palliative, and include trabeculectomy, and drugs able to decrease the intraocular fluid accumulation.

However, AMD is the most common eye disease, and it covers different types of lesions affecting the macula. Its early-stage, so-called “early AMD”, is usually asymptomatic but could evolve into two advanced forms: the atrophic (dry) or the exudative (neovascular or wet) AMD. These aggravations can occur over few months or several years, and depend on various and poorly understood parameters, such as demographic, environmental, genetic, gender, and phenotypic risk factors [8]. Contrary to early AMD, the advanced forms are characterized by a central vision loss combined with metamorphopsia (distorted perception of straight lines and wavy images) and with scotoma (dark spot), rendering progressively the daily activities (e.g., reading and driving) tricky to impossible (Figure 3) [4]. The mechanisms underlying these two advanced forms are not entirely characterized, which make the development of a global cure against AMD very challenging. To this day, no efficient and curative treatment exists for AMD, and this pathology is considered in western countries as a leading cause of blindness for patients over 65 years. 

## 2. Pathogenesis, Development, and Characteristics of AMD

### 2.1. Epidemiology and Diagnosis

In 2015, around 67 million people in the EU were affected by AMD with severe visual loss [9]. This number is expected to increase by 15% by 2050, due to the increase in life expectancy [10]. The annual incidence for late AMD stage is 0.5 cases per 1000 individuals under 70, and 6.7 cases per 1000 individuals over 70 years old (Figure 4 and Figure 5); in addition, this prevalence is higher in women than in men [11] (Figure 4) [11]. 

Several epidemiological studies have demonstrated a strong correlation between the prevalence of AMD and the age of patients: (i) BEAVER DAM, an American study (1988–1990, 4926 participants) [12], (ii) BLUE MOUNTAINS, an Australian study (1992–1994, 2454 participants) [13], and (iii) EUREYE, a European study (2000–2003, 5040 participants) [14]. 

More recently, in 2021, a review referenced thirty studies, including eleven after the 2000s, on four continents (Europe, Asia, Oceania, and Northern America) [15]. This work pointed out an annual incidence increase correlated with age, and higher in Asia and Oceania. 

The diagnosis of AMD is currently based on several visual tests and on multimodal imagery. A very common and convenient test is the Amsler grid (or Amsler chart), developed by Marc Amsler (Zurich, Switzerland) in the 1940s, which allows a self-diagnosis “at-home”. This test consists in measuring the visual perception of a grid, the central part of which may appear wavy, irregular, or distorted in patients with macular diseases. The visual acuity can also be evaluated using the ETDRS scale score (Early Treatment Diabetic Retinopathy Study) [16]. Nevertheless, eye imagery remains the most precise method to diagnose the pathology and to precisely characterize its form and evolution. In particular, ophthalmologists can monitor the retina, retinal pigment epithelium (RPE), and choroid morphological changes by fundus exam, color fundus photography [17], fundus autofluorescence (FAF) [18], optical coherence tomography (OCT) [19], infrared reflectance (IR) [20], fluorescein/indocyanine green angiographies, or OCT angiography. These various ocular-imaging techniques allow a classification of the different AMD forms (Figure 6) [21].

### 2.2. Age-Related Maculopathy (ARM) and Its Progression to AMD Advanced Stages

ARM is characterized by the emergence and the progressive growing of lipidic aggregates, organic wastes, and β-amyloid peptides named drusens (Figure 6A–C). They are rejected by cells via exocytosis and accumulated in the retinal and subretinal spaces. In the early ARM stage, drusens’ diameters do not exceed 63 µm; their growth is combined with the production of inflammatory factors (cytokines). At this stage, the macula’s lesions remain asymptomatic or would lead to a slight decrease in visual capacities [22]. 

Some aggravating factors responsible for irreversible ARM evolution to AMD have been identified; however, not entirely understood. In addition to the above-mentioned age and gender [12,13,14], the patient’s lifestyle (smoking, diet and body mass index, and education) seems to play a role in the disease’s evolution. Some genetic factors have also been underlined, such as gene mutations coding for the complement factor, as well as phenotypic factors, including pigment abnormalities. Of note, biochemical markers (high-density lipoprotein cholesterol (HDL-C), docosahexaenoic acid (DHA), eicosapentaenoic acid (EPA), zeaxanthin, or lutein) are currently not considered in the predictive models but they might be studied as potential markers in the future, as they are easily accessible via blood analysis [8].

The most common marker of the evolution to AMD stages is the presence of larger drusens, with diameters up to 125 µm [23,24]. Importantly, the two advanced AMD stages characteristics are very distinct. The exudative form is characterized by an abnormal neovascularization invading choroid, Bruch’s membrane and RPE, which leads to an accumulation of fluid (Figure 7). Contrarily, the atrophic (dry) form presents no new vessels formation [25,26]. It should be noted that both advanced forms are not mutually exclusive: the atrophic form can eventually develop neovascularization and switch to the exudative form, and patients with exudative form may display some atrophy after few years [27]. 

In contrast with ARM, the late AMD stages are symptomatic; they are characterized by a vision decrease as well as a progressive and irreversible loss of the central vision. These symptoms are due to several factors including (i) geographic atrophy (GA), and/or (ii) subretinal hemorrhage, in the exudative form, induced by the invasion of RPE and/or retina by abnormal blood vessels. The exudative form that is more severe than the atrophic one affects 10% to 15% of patients causing 90% of acute blindness [28].

### 2.3. AMD Characteristics

#### 2.3.1. Common Characteristics of the AMD Advanced Stages

Both atrophic and exudative AMD forms are characterized by chronic inflammation, which may be a key player in AMD evolution through different pathways. Inflammation induces endothelial dysfunction in choroidal vessels, development of basal deposits and drusens, and degeneration of Bruch’s membrane. 

Some other factors, such as β-amyloids leading to drusen accumulation, and reactive oxygen species (ROS) are involved in the aggravation of the two advanced forms, even if their precise roles remain elusive and must be investigated (Figure 8) [29].

#### 2.3.2. Clinical Specificities of the Atrophic AMD form (Dry-AMD)

In the atrophic form, the drusen accumulation that leads to the thickening of the Bruch’s membrane (BrM) is a key feature. The expansion of drusens results in RPE cells dysfunction and may lead to their death. Drusens are also responsible of the photoreceptors degeneration and death as well as the alteration of the RPE fluid efflux through the BrM, choriocapillaris, and peripapillary atrophy (Figure 6D and Figure 9) [30,31,32,33,34]. Finally, drusens also induce and sustain eye inflammation. 

In addition, in the elderly, biochemical and anatomical changes occur in the BrM resulting in a decrease in the nutrients’ flow through BrM [35]. These changes could lead to the aggravation of AMD symptoms. Consequently, hypopigmentation areas of the RPE monolayer are observed in the macula and are also associated with mild to moderate retinal degeneration and vision loss. 

Today, no therapeutic options can be proposed to patients to slow down the atrophic AMD progression. It should be noted that, in some cases, it may switch to the exudative form, and this must be prevented [36]. In line with this consideration, several strategies of care are currently under investigation to decrease drusens’ formation of and avoid RPE disorders. Nevertheless, the lack of the global comprehension in the AMD’s aggravation mechanisms impairs drastically the emergence of a rational and efficient treatment. 

The only current therapies against atrophic AMD are symptomatic, and they aim at (i) regulating the visual cycle to lower drusens formation [29], (ii) counteracting the choroid’s atrophy and choriocapillaries’ (CC) atrophy [37,38], (iii) avoiding the death of the photoreceptor and the RPE cells, (iv) reducing the oxidative stress in the eye [39], and finally (v) reducing eye inflammation [40,41] (Figure 9). Some of these treatments have been tested in clinical trials (phase 1, 2, and 3). These different approaches are presented in the Section 3 of this review.

#### 2.3.3. Clinical Specificities of the Exudative AMD form (Wet-AMD)

The exudative form is characterized by an invasive neovascularization of the choroid, Bruch’s membrane, RPE and retina, which contributes, in combination with GA, to a progressive vision loss. This neovascularization [42] is mediated by the overexpression of the vascular endothelial growth factor (VEGF) as well as by local inflammatory cytokines, such as tumor necrosis factor-alpha (TNF-alpha) (Figure 10) [43,44,45]. Indeed, in healthy eyes, VEGF plays a physiological role in the development and trophic maintenance of the choriocapillaries [46]. VEGF also protects retinal neurons from apoptosis in ischemic conditions. Conversely, its pathological overexpression induces the formation of angiogenic germs in the fovea, a small depression of the macula that contains the highest concentration of photoreceptors. VEGF overexpression also leads to activation, survival and proliferation of new vessels (CNV) commonly divided into three types: type 1 to type 3 new vessels. 

These CNV may be categorized into three major types depending on their localization: **Extrafoveal**, when neovascularization is located between 200 µm and 2500 µm from the geometric center of the foveal avascular zone,**Juxtafoveal**, when neovascularization is restricted to an area up to 199 µm from the geometric center of the foveal avascular zone (this area may include portions of the foveal avascular zone),**Subfoveal**, when neovascularization is directly beneath the geometric center of the foveal avascular zone.

New vessels progressively invade the BrM, the pigment epithelium and/or the sub-retinal space under the macula, causing significant retinal detachment. In addition, these new vessels are characterized by an abnormal structuration allowing lipids, plasma and blood extravasation, resulting in fluid accumulation in the invaded tissues (Figure 6E–G). This fluid retention induces a drastic aggravation of the retinal detachment and subretinal hemorrhages causing degeneration and death of the photoreceptors. These morphological changes quickly damage the macula and dramatically impair the visual acuity. The ultimate clinical evolution is characterized by the appearance of a fibrous scar, called disciform [4]. 

The exudative AMD form can develop suddenly, leading in few weeks or months to photoreceptors’ death, to severe decrease in VA and to permanent central scotoma. If untreated, exudative AMD progresses irreversibly to central blindness [42]. Nevertheless, some therapeutic options emerged since the 1980s. These strategies consist in tackling the neovascularization, either directly, using laser beam or photodynamic therapy, or indirectly, using anti-angiogenic drugs (anti-VEGF agents) [47]. However, these approaches are currently counterbalanced by several recent studies suggesting that, alone, VEGF is not sufficient to significantly increase choroid vascularization. Combined activation or inhibition of various angiogenic factors may lead to the pathological vascularization. These redundancy in pro-angiogenic pathways underline the need of targeting other receptors/endothelial factors to obtain a satisfactory response [48]. The overall experience concerning the injection of anti-VEGF demonstrated their good tolerability; however, a controversy exists on their potential atrophic action in the long term. In any case, even if anti-angiogenic treatments offer short-term clinical benefits, this treatment is not curative, it may only delay or pause the progression of the disease. Therefore, great efforts, in academia as well as in the industry, are currently dedicated to identifying alternative strategies. These are presented and discussed in the Section 4 of this review.

## 3. Ongoing Research and Trials in Therapeutic Options against the Atrophic form of AMD (Dry-AMD)

The search for therapeutic options towards atrophic AMD has started less than two decades ago but it is still an active field of research (Figure 9 and Figure 11). A particular emphasis has been given to the drugs formulation, for patients’ long-term acceptance and compliance with the treatment. In particular, intraocular injections may be considered as traumatic and dissuasive by some patients. Systemic treatments or eye drops, more difficult to develop, might be clearly preferred even if they do not allow a precise control in the amount of drug in the local targets, and may lead in some cases to systemic adverse effects (in the case or systemic drugs).

### 3.1. Visual Cycle

Vision consists in the conversion of light into a nervous influx, which is transmitted to the brain through the optic nerve. When light reaches the photoreceptors, retinal 11-*cis*, a vitamin A derivative covalently bound to rhodopsin, undergoes a photochemical reaction and is converted into its more stable all-*trans* isomer. This stereochemical change in the retinal structure induces a conformational change in the rhodopsin structure, resulting in the activation of the nervous influx.

The visual cycle consists in the regeneration of the retinal-11-cis from its all-*trans* isomer, thanks to a succession of enzymatic reactions occurring in the photoreceptor cells and in the RPE cells. At first, all-*trans* retinal is reduced into all-*trans* retinol (vitamine A) in photoreceptor cells. Then all-*trans* retinol migrates to the RPE, and undergoes an esterification mediated by lecithin retinol acyltransferase (LRAT). Subsequently, all-*trans* retinyl ester is isomerized, then hydrolyzed by retinal pigment epithelium-specific 65 (RPE65) to form 11-*cis* retinol. Finally, 11-*cis* retinol is oxidized into 11-*cis* retinal by 11-*cis* retinol dehydrogenase (RDH), before returning into the photoreceptor cells [49,50]. 

The precise turn-over of the visual cycle is deeply altered in atrophic AMD patients. Thus, in RPE cells, all-*trans* retinal isomer dimerizes to afford *N*-retinylidene-*N*-retinylethanolamine (A2E). A2E could form aggregates leading to lipofuscin, a key aggravating factor (Figure 12 and Figure 13) [50,51,52]. In line with these observations, the deceleration of the visual cycle turn-over has been proposed as a therapeutic option [51]. To date, five small organic molecules have been developed to interfere with this mechanism (Table 1). All of them are administered via oral pills for systemic exposure. 

Emixustat or ACU-4429

Emixustat (Acucela, Inc., Seattle, WA, USA; Kubota Vision Inc., Tokyo, Japan; Otsuka Pharmaceutical Co., Ltd., Tokyo, Japan, 2012) is the first small-sized modulator of the visual cycle [53,54]. This molecule inhibits the conversion of all-*trans* retinyl ester into 11-*cis*-retinol, catalyzed by RPE65. Emixustat binds the retinoid site of RPE65 and inhibits its activity via its hydroxyl group, (IC_50_ = 4.4 nM) [55]. In turn, emixustat induces the deceleration of the visual cycle and the reduction in A2E accumulation. Eximustat is a chiral derivative and its (*R*) enantiomer is the most potent one [56]. Phase 1 and phase 2 studies (NCT02130531, 2014–2016; NCT01002950, 2009–2014) [57,58] have been performed to evaluate the safety, the tolerability and the pharmacokinetics/pharmacodynamics of emixustat at different doses for 90 days (2, 5, 7 or 10 mg). The drug was well tolerated upon daily administration for 2 weeks (phase 1 trial), and [54] at this level emixustat displayed a reversible dose-response effect (phase 2 trial). Two doses were interrupted (7 and 10 mg) due to side effects, even if in the 5 and 7 mg groups, two patients (in each group) showed a decrease in VA compared to the placebo patients [59]. Of note, this molecule has been described, in in vivo studies, for the treatment of Stargardt macular dystrophy [60], a pathology in which lipofuscin formation is also observed. Emixustat dose-effects have been evaluated in a phase 2/3 study (SEATTLE, NCT01802866, 2013–2017) [61,62] with different drug amounts (2.5 to 10 mg) for a treatment of 24 months. However, whatever the dose was, the growth rate of GA was not reduced by the drug.

CU239

More recently, in 2018, a new compound CU239 has been identified to inhibit RPE65 (IC_50_ = 6 μM) as emixustat [63]. CU239 inhibits the RPE65 isomerase activity by competing with all-*trans* retinyl ester that may cause retinal degeneration [63].

Fenretinide

The interaction between transthyretin (TTR) and retinol 4 (RBP4) is responsible for all-*trans* retinol transport from the photoreceptors to the RPE cells, [64,65] is targeted by Fenretinide, a RBP4 antagonist that mimics vitamin A [66]. Fenretinide inhibits the TTR-RBP4 complex formation (IC_50_ = 56 nM) [67,68]. A phase 2 study (NCT00429936, 2007–2010) [69] evaluated the efficacy of this molecule (dose given to patients: 100 mg over 24 months) in atrophic AMD. This trial showed a reduction in the growth of GA lesions and a decrease in neovascularization. This result suggests that fenretinide may prevent the evolution of the AMD from its dry to its wet form [66].

A1120

A1120 is a small molecule with high affinity for RBP4 (K_i_ = 8.3 nM) and blocks the interaction between RBP4 and TTR. A1120 inhibits the TTR-RBP4 complex formation (IC_50_ = 14.8 nM) [67]. It induces retinol conformational changes and leads to the blockade of lipofuscin formation in an Abca4 mouse model [67]. Remarkably, the scaffold of this molecule differs strongly from vitamin A, suggesting an original mode of interaction with its target [67]. However, to date, this molecule has not reached the clinical trial stage. Due to poor human liver microsome (HLM) stability of A1120 (∼3%, 30 min incubation), several new antagonists have been designed [70]: (i) by replacing the aryl carboxylic acid by its isosters, (ii) to increase flexibility, by replacing piperidine ring with an acyclic *N*-methyl-*N*-(2-phenoxyethyl)amido linker, (iii) finally, by replacing piperidine ring by bicyclic fragments. This last bispecific analogues research highlighted an isostere of A1120, (±)-1-(4-(2-(trifluoromethyl)phenyl)piperidine-1-carbonyl)pyrrolidine-2-carboxylic acid as promising oral treatment for atrophic AMD [71,72].

ALK-001

ALK-001, or C20-D3-vitamin A, is a deuterated derivative of vitamin A, which prevents A2E formation [73]. A pre-clinical study revealed that ALK-001 uses a physical-chemistry property, the effect of deuteration, in order to profoundly modulate the vitamin A dimerization process by slowing it down (Figure 1) [52]. ALK-001 was developed by Alkeus Pharmaceuticals, Inc. (Boston, MA, USA) to slow down or stop vision loss in atrophic AMD and Stargardt disease [52]. After a phase 1 study (NCT02230228, 2014–2015) [74] to evaluate its safety and pharmacokinetics in 40 healthy patients, a phase 3 study (SAGA, NCT03845582, 2019–2023) [75] recently recruited 300 AMD patients to evaluate its potential benefits.

**Table 1 molecules-27-05089-t001:** Drugs targeting the visual cycle.

Drugs	Structure	Target	Clinic or Research	Formulation	Ref.
Emixustat	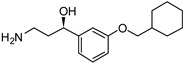	RPE65	1, 2	Oral (Tablet)	NCT01802866 [61] NCT02130531 [57] NCT01002950 [58]
CU239	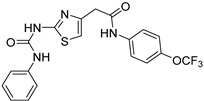	RPE65	R	-	[63]
Fenretinide	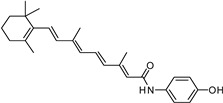	RBP4	2	Oral capsules	NCT00429936 [69]
A1120	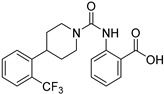	RBP4	R	-	[67]
ALK-001	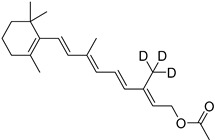	Vit. A	1	Oral capsules	NCT02230228 [74] NCT03845582 [75]

### 3.2. β-Amyloid (Aβ)

The β-amyloid (Aβ) accumulation in drusen, in particular the amyloid peptides Aβ1–40 (Aβ40) and Aβ1–42 (Aβ42), is responsible for AMD aggravation [29,76]. In addition, Aβ inhibits the complement factor I (CFI) activity, responsible for the alternative complement cascade, and participates therefore in the inflammatory induction [77]. Thus, lowering Aβ production might be a therapeutic option for atrophic AMD treatment. 

RN6G

RN6G (also called PF-04382923, Table 2) is an anti-amyloid-β monoclonal antibody developed by Pfizer (New York, NY, USA), which binds and traps Aβ in the retinal periphery. In mice, it reduces Aβ toxic accumulation in the macula [76,78]. Two phases 1 studies (NCT01003691, 2009–2013; NCT00877032, 2009–2015) [79,80] have been completed to assess the RN6G optimal dose; however, in both cases, no improvement in visual acuity (BCVA assays) has been reported. In addition, a phase 2 study (NCT01577381, 2012–2016) [81] showed that there were neither improvements in BCVA nor in ocular lesions (GA). 

GSK933776

GSK933776 (Table 2) is a fully humanized mouse anti-human Aβ immunoglobulin G1, which binds the Aβ N-terminal [82]. This drug, developed by GlaxoSmithKline (Brentford, UK), was designed to restore the CFI bioactivity, and it has been proved to be active in vivo [77]. GSK933776 was initially used in the treatment of the Alzheimer’s disease. For the treatment of AMD, two trials have been completed: a phase 1 study (NCT02033668, 2014–2017) [83] to evaluate the pharmacokinetics and different formulation modes (intravenous vs. subcutaneous injection, dose: 200 mg) and a phase 2 study (NCT01342926, 2011–2017) [84]. This latter showed no improvement in GA enlargement rate and VA. Moreover, no correlation was observed between GA enlargement rate and the CFI variations [85]. 

**Table 2 molecules-27-05089-t002:** Drugs targeting β-Amyloid.

Drugs	Structure	Clinic or Research	Formulation	Ref.
RNG6	Monoclonal antibody	1	Intravenous	NCT01003691 [79]
				NCT00877032 [80]
		2		NCT01577381 [81]
GSK933776	Monoclonal antibody	1	Intravenous or subcutaneous injection	NCT02033668 [83]
		2		NCT01342926 [84]

### 3.3. Choriocapillaries (CC) Atrophy

CC ensures nutriments intake and waste removal in the RPE [86]; their alteration induces ocular hypoxia, which is potentially responsible for drusens’ multiplication. As a result, vasodilators have been proposed to counteract CC atrophy [37,38]. 

MC-1101

Hydralazine (so-called MC-1101, or Apresoline^^®^^ hydrochloride; MacuCLEAR Inc., Richardson, TX, USA) (Table 3) [87] is an antihypertensive, anti-inflammatory, antioxidant and vasodilating agent, approved in 1997 (FDA) to regulate blood pressure [88]. On animal models (rats and rabbits) MC-1101 showed an increase in the choroidal blood flow in the macula [39,89]. Furthermore, hydralazine exerts an antioxidant action and reduces atrophic AMD lesions [89]. In humans, two phase 1 studies (NCT01013376, 2009; NCT01922128, 2013–2014) [90,91] and a phase 2/3 study (NCT01601483, 2012–2014) [92] dealing with hydralazine safety profile (including atrophic AMD patients) and its optimal regimen (dose and administration mode) have been implemented. These assays proved that no adverse effects (such as cardiovascular effects, ocular toxicity or degradation of the blood–eye barrier) occurred when the drug was administered topically (eye drops, 1% in an ophthalmic solution). 

Moxaverine

Another vasodilatator assayed is moxaverine (Kollateral forte^^®^^), a papaverine derivative developed by Ursapharm (Saarbrücken, Germany) (Table 3) [93]. This phosphodiesterase inhibitor was initially used for the treatment of peripheral microcirculatory impairment [94]. In atrophic AMD, moxaverine has been assayed in phase 2/3 clinical trial (NCT00709449, 2008–2009) [95] with a 150 mg dose as intravenous infusion. In this assay, an increase in the choroidal blood flow has been observed in patients suffering AMD and glaucoma, [93] revealing its potential efficiency. These results should be confirmed by additional clinical trials. 

Sildenafil

Sildenafil citrate (Viagra^^®^^) (Table 3), developed by Pfizer (New York, NY, USA), is a phosphodiesterase type-5 inhibitor marketed to treat the male erectile dysfunction [96]. A “pilot” phase 2 study (NCT01830790, 2013–2015) [97] enrolling only 10 AMD patients (exudative and atrophic) underlined an increase in the choroid thickness (dose: 100 mg) in both AMD forms) [98]. However, because of insufficient support to complete the recruitment, this trial was prematurely terminated. In 2019, another study with 23 subjects, including 15 AMD patients, showed an increase in choroid thickness after administration of sildenafil (oral dose: 100 mg). Nevertheless, sildenafil may induce a weaker vascular response in older patients [99]. More studies should be implemented to confirm these preliminary results. 

Trimetazidine

Trimetazidine (TMZ), Vastarel^^®^^ (Table 3), is a piperazine derivative developed by Servier (Suresnes, France) to inhibit fatty acids and glucose oxidation [100]. This drug has been initially used for its anti-ischemic properties (1981) [101]. However, TMZ may also prevent microvascular abnormalities in the choroid and retina. These protecting effects have been validated in glaucoma and degenerative myopia. TMZ improves the sensitivity to contrast and the patient’s visual acuity (treatment of 20 mg TMZ, twice a day for 6 months) [102]. Nonetheless, a phase 3 study that was carried out for exudative and atrophic AMD patients (dose of 35 mg, twice a day, over 3–5 years), failed to demonstrate a clinical benefit on CNV [103], even if a possible preventive effect in delaying atrophy has been suggested.

**Table 3 molecules-27-05089-t003:** Drugs targeting choriocapillaris atrophy.

Drugs	Structure	Clinic or Research	Formulation	Ref.
MC-1101	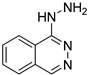	1	Eye drops	NCT01013376 [90]
2/3	NCT01601483 [92]
1	NCT01922128 [91]
Moxaverine	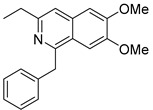	2/3	Intravenous infusion	NCT00709449 [95]
Sildenafil	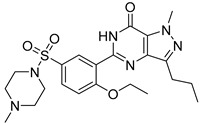	2	Oral	NCT01830790 [97]
Trimetazidine	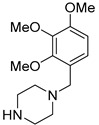	3	Oral	ISRCTN99532788 [103]

### 3.4. Oxidative Stress

Mitochondrial damages and oxidative stress in RPE are key players in AMD, even if these mechanisms have not been understood yet [104]. Indeed, RPE cells are exposed to chronic oxidative stress due to their oxygen requirements and their exposure to peroxilated lipids [105]. At the RPE cells’ level, a set of ROS and free radicals are produced, such as superoxide, hydrogen peroxide and hydroxyl radicals [106]. This accumulation is also associated with lipofuscin and β-amyloid overproductions [107]. Damages to the RPE may lead to the photoreceptors’ apoptosis. Furthermore, these oxidative stresses could aggravate the inflammatory context. Moreover, inflammatory stimuli increase the production of reactive oxygen intermediates and reduce the bioavailability of antioxidants, initiating thereby a vicious circle [108]. To summarize, oxidative stresses lead to the aggravation of the disease, and their downregulation may lead to relevant therapeutic options (Figure 14) [109]. Thus, compounds with antioxidant properties or aimed to interfere with ROS production have been proposed as potential treatments for patients with AMD.

#### 3.4.1. Small Molecules

OT-551

OT-551 (Table 4), developed by Othera Pharmaceuticals (Exton, PA, USA), is a derivative of the well-known radicals’ quencher TEMPO. OT-551 is namely a lipophilic prodrug of tempol (TP). Delivered topically by eye drops, OT-551 has the remarkable capacity to penetrate the cornea towards the posterior segment of the eye to reach the macula. The surrounding esterases then convert OT-551 into its corresponding alcohol TP-H (Figure 2) making it an interesting candidate for AMD treatment [110]. A non-direct mechanism of action of OT-551 consists of downregulating the Nuclear factor E2-related factor (Nrf-2), overexpressed in inflammatory diseases. In consequence, OT-551 may act as an antioxidant and anti-inflammatory factor [110].

A phase 2 study (NCT00306488, 2006–2011) [111], with 11 participants, reported an improvement in BCVA for patients treated with OT-511 (0.45% concentration eye drops, three times a day, for 24 months). In addition, the prodrug seemed to be well-tolerated by patients. However, these positive results have been counterbalanced by the absence of improvements for other AMD markers (lesions size, retinal sensitivity and drusen area) [112]. Moreover, a larger phase 2 study (OMEGA, NCT00485394, 2007–2010) [113], did not confirm the VA improvement (198 participants, 18 months, same dose and administration mode) [114]. These discrepant studies may suggest that OT-551 is not an adequate treatment.

Risuteganib

Risuteganib (ALG-1001, Luminate^^®^^) (Table 4), developed by Allegro Ophthalmics (San Juan Capistrano, CA, USA), is a small pseudo peptide targeting the integrin heterodimers (α_V_β_3_, α_V_β_5_, α_5_β_1_ and α_M_β_2_) involved in angiogenesis, vascular leakage and inflammation. A phase 2 study (NCT03626636, 2018–2019) [115] evaluated the safety profile and the efficacy of risuteganib in atrophic AMD patients. It concluded to an improvement in BCVA, characterized by a gain of Early Treatment Diabetic Retinopathy Study (ETDRS) letters; however, no longer-term studies have been reported [116].

**Table 4 molecules-27-05089-t004:** Drugs targeting oxidative stress.

Drugs	Structure	Target	Clinic or Research	Formulation	Ref.
OT-551	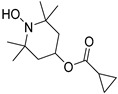	Nrf-2, ROS	2	Eye drops	NCT00485394 [113] NCT00306488 [111]
Risuteganib	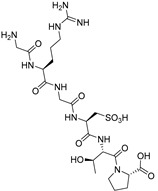	integrin heterodimers	2	Intravitreal injection	NCT03626636 [115]

#### 3.4.2. Nutritional Supplements

Changes in lifestyle and nutritional supplements have also been proposed to slow down AMD progression [117]. Thus, two studies dealing with the use of nutritional supplements (vitamins C, β-carotene and minerals such as zinc and copper) have been carried out by Bausch and Lomb, Inc. (Laval, QC, Canada): Age-Related Eye Disease Study 1 (AREDS1) [118] and Age-Related Eye Disease Study 2 (AREDS2) [119]. These nutrients downregulate AMD progression (20% with antioxidant vs. 28% without) and vision impairment (19%) after 5 years [120]. Another study (TOZTAL, 2007) consisting in giving taurine, omega-3 fatty acids, zinc and lutein supplements (Table 5) showed no changes in VA [121]. Other dietary supplements, such as curcumin [122,123,124] or resveratrol [125], have also been studied, but no conclusive results have been reported yet (Table 5). However, by comparing several epidemiological studies, a risk reduction in developing the AMD late form has been observed by taking carotenoids (lutein, zeaxanthin, and meso-zeaxanthin) (Table 5) [126]. Thanks to their antioxidant activity and anti-inflammatory position, the carotenoids act as a neuroprotector and their use may thereby be considered as a nutraceutical strategy to prevent AMD aggravation. To conclude, nutrient supplementation is a part of research in the prevention of AMD, but further studies are needed to demonstrate their efficacy [127].

#### 3.4.3. Neuroprotection

Neuroprotective factors protect RPE and photoreceptors cells from the oxidative stress damages [39]. 

CNTF

The ciliary neurotrophic factor (CNTF) (Table 6), developed by Neurotech Pharmaceuticals (Cumberland, RI, USA), is an IL-6 type cytokine. In a set of neurodegenerative disorders, CNTF is reported to delay the symptoms’ aggravation. In the specific case of ocular diseases, it improves photoreceptors and RPE cells survival. In animals suffering retinitis pigmentosa, CNTF proved to slow down the retinal degeneration [128,129]. In humans, CNTF is secreted by modified human retinal pigment epithelium cells, trapped in a polymer implant (NT-501) which is surgically grafted into the vitreous body [130]. For patients suffering from atrophic AMD, a phase 2 study (NCT00447954, 2007–2016) [131] evaluated the safety and efficacy of CNTF in this implant. The encapsulated cells remain active after 24 months, and positive pharmacokinetic results with continuous delivery of CNTF have been observed.

Brimonidine Tartrate

Brimonidine tartrate (Alphagan^^®^^) (Table 6), developed by Allergan, Inc. (Dublin, Ireland), is a small-sized α2-adrenergic receptor agonist [78], used for the treatment of ocular hypertension and glaucoma (FDA approved in 1996) [132,133]. Brimonidine is reported to reduce intraocular pressure (IOP) [134]. This drug prevents RPE and photoreceptors’ apoptosis, by allowing the release of neutrophins such as brain-derived neurotrophic factor (BDNF), CNTF and basic fibroblast growth factor (b-FGF) [135]. 

This molecule was administered to patients as an intravitreal biodegradable implant for clinical studies. Allergan performed two phase 2 studies (NCT00658619, 2008–2018; BEACON, NCT02087085, 2014–2019) [136,137] to evaluate its safety and efficacy. Thus, injecting 200 or 400 µg of brimonidine for 24 months demonstrated a change in the size of the lesions (smaller GA lesions), especially for patients with the largest ones [138]. In the BEACON study, an intravitreal implant of brimonidine (400 µg) has been applicated every three months up to the 21st month, showing a reduction in the GA lesions after 30 months [139].

Tandospirone

Tandospirone (AL-8309B, Sediel^^®^^) (Table 6), developed by Alcon Laboratories, Inc. (Geneva, Switzerland), is a 1A serotonin agonist [39] used as antioxidant and anti-depressant. In the external retina, this molecule slows down the activation of microglia and the deposition of different complement proteins such as C3, factor B, factor H and MAC [140]. It preserves RPE cells and photoreceptors from oxidative stress and prevents retinal cell apoptosis [78,140]. Tandospirone can be administrated through eye drops. 

A phase 3 study (GATE, NCT00890097, 2009–2014) [141] did not point out any safety concerns for its clinical use, but no change in the GA lesion’s growth was reported compared to untreated patients [142].

**Table 6 molecules-27-05089-t006:** Drugs targeting neuroprotection.

Drugs	Structure	Target	Clinic or Research	Formulation	Ref.
CNTF	Protein	Photo and RPE cells	2	intravitreal injections (Implant: NT-501)	NCT00447954 [131]
Brimonidine	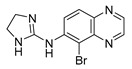	Photo and RPE cells	2	intravitreal injections (Implant)	NCT00804921 [143] NCT00864838 [144] NCT00658619 [136] NCT02087085 [137]
Tandospirone	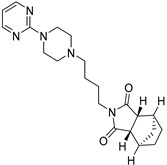	Photo and RPE cells	3	Eye drops	NCT00890097 [141]

### 3.5. Inflammatory Pathways

Many daily stressors, such as UV radiation, oxidation and infection result in a chronic low-grade inflammation, which is a relevant parameter for several age-related degenerative diseases [145,146], including AMD [147]. In AMD, chronic inflammation may induce endothelial dysfunction in choroidal vessels, development of basal deposit and drusens as well as degeneration of Bruch’s membrane. In addition, AMD is associated with an increase in the vessel wall’s permeability, a leakage into the surrounded interstitial tissue and edema, an activation of blood-borne inflammatory cells and an involvement of the complement system, which all are conventional inflammation hallmarks [148]. Thus, various drugs have been studied to target inflammation (Table 7).

#### 3.5.1. Complement Cascade

The complement system is a group of proteins which stimulate inflammation and opsonization during the immune response. Three distinct biochemical pathways leading to cell swelling [149] exist: classical, alternative and lectin pathway (Figure 15). The terminal cell lysis is induced by the formation of the membrane attack complex (MAC). MAC also acts at the level of the CC loss and, possibly, for drusens formation [150]. In atrophic AMD, the alternative pathway of the complement cascade may be therefore targeted. 

Lampalizumab

Lampalizumab (FCFD4514S) developed by Genentech, Inc. (South San Francisco, CA, USA) from Hoffmann-LaRoche (Basel, Switzerland) is an antigen-binding fragment (Fab) from monoclonal antibody with an anti-inflammatory activity that targets the factor D. Two phase 2 studies (MAHALO, NCT01229215, 2010–2016; NCT02288559, 2014–2019) [151,152] evaluated its safety: the drug was given by intravitreal injection during 18 months and 24 weeks at a 10 mg dose. Great benefits for patients have been reported, with a 20% average reduction in the lesion’s progression, which reached 44% with a subgroup of CFI risk-allele carriers [153]. In addition, the lesions of treated patients suffering from outer retinal tubulation (ORT) grow slower compared to untreated patients [154]. An extension of the precedent studies (NCT01602120, 2012–2019) [155] was performed but it was terminated prematurely because of a lack of efficacy. In addition, phase 3 clinical trial studies (CHROMA, NCT02247479, 2014–2019; SPECTRI, NCT02247531, 2014–2019) [156,157] were performed to evaluate the efficacy and safety of intravitreal lampalizumab injection (10 mg) for 96 weeks. An extension of the phase 3 study (OMASPECT, NCT02745119, 2016–2019) [158] was terminated in 2019 and tarnished the former results since no significant reduction in geographic atrophy has been observed in comparison to a placebo group. Therefore, Roche, announced the end of the study in 2017 [159]. 

Danicopan

Danicopan (ACH-4471, ACH-044471, ALXN2040) is an oral factor D inhibitor (IC_50_ = 5.8 nM), preventing alternative pathway C3 convertase formation [160]. Initially, danicopan was used to block the alternative pathway for paroxysmal nocturnal hemoglobinuria. Thanks to factor D, the intravascular hemolysis was controlled, and the extravascular hemolysis mediated by C3 was anticipated [161]. A phase 2 study (NCT05019521, 2021–2025) [162] has been initiated to evaluate the safety, efficacy and pharmacokinetics of danicopan with multiple doses on 330 participants with GA AMD.

POT-4 and APL-2

**POT-4 or AL-78898A**, developed by Potentia Pharmaceuticals (Crestwood, KY, USA), is a 13 amino acids cyclic peptide derived from compstatin, a C3-targeted complement inhibitor. This molecule has been reported to inhibit the conversion of C3 to C3a and C3b, preventing in turn MAC formation [163]. A phase 2 study (NCT01603043, 2012–2014) [164] reported the formation of product’s deposits for four patients among seven (57.4%) due to its poor solubility [165]. **APL-2** (pegcetacoplan) is a synthetic cyclic peptide conjugated to a PEG polymer developed by Apellis Pharmaceuticals (Waltham, MA, USA). This molecule has been developed to counteract solubility concerns, and it corresponds to a 40000 Da PEGylated conjugate of POT-4, which is also able to block the C3 complement protein [166]. A phase 2 study (FILLY, NCT02503332, 2015–2019) [167] was completed to evaluate the safety and tolerability of APL-2 and to demonstrate the relevance of multiple intravitreal injections (every month or every two months). This study evidenced a reduction in the growth of GA lesions [166,168]. Other studies are currently ongoing: a phase 1 study (NCT03777332, 2018–2021) [169] and three phase 3 studies (OAKS, NCT03525600, DERBY, NCT03525613, 2018–2023; GALE, NCT04770545, 2021–2025) [170,171,172]. The OAKS study showed a significant reduction in the GA lesion growth (22 and 16%) after 12 months but, for the DERBY study, the reduction was less important with 11 and 12% after 12 months [173]. Moreover, both studies demonstrated a favorable safety profile which highlights APL-2 as very promising treatment for geographic AMD. Finally, the GALE study is an extension of FILLY, OAKS and DERBY studies on the participants who have completed them. This 36-months study aims to evaluate the efficacy of APL-2 in geographic atrophy secondary to AMD subjects.

Eculizumab

Eculizumab (Soliris^^®^^), developed by Alexion Pharmaceuticals (Boston, MA, USA), is a monoclonal antibody that targets the C5 complement protein and prevents its conversion into C5a and C5b. Eculizumab has been approved by FDA in 2007 to treat atypical hemolytic uremic syndrome and paroxysmal nocturnal hemoglobinuria [174]. A phase 2 study (COMPLETE, NCT00935883, 2009–2017) [175,176,177] evaluated the effects of intravenous injection of eculizumab against atrophic AMD (600 mg or 900 mg). Normal-luminance and low-luminance visual acuities were analyzed through this study, without observing any significant results on the GA’s growth rates or on the drusen volume reduction.

Tesidolumab

Tesidolumab (LFG316) is a fully human IgG1 antibody developed by Novartis (Basel, Switzerland), that targets the complement protein C5 [166]. A phase 1 study (NCT01255462, 2010–2012) [178] showed the safety and tolerability of tesidolumab (doses up to 5 mg) in patients suffering of advanced AMDs. Next, two phase 2 studies (NCT01527500, 2012–2019; NCT02515942, 2015–2019) [179,180] evaluated tesidolumab multiple-dose intravitreal injections (12 injections every 28 days). The second phase 2 study was in combination with CLG561, an antibody complements pathway inhibitor. However, a little gain was observed in VA but no improvement in GA lesion size was reported. 

CLG561

CLG561 is a fully human antibody Fab properdin inhibitor. Properdin is a plasma glycoprotein acting as a positive regulator of the alternative pathway allowing C3a/C3b formation then C5a/C5b formation [181]. In addition to a phase 2 study (NCT02515942, 2015–2019) [180], a phase 1 study (NCT01835015, 2013–2014) [182] showed the safety and tolerability of CLG561 in AMD patients after 84 days for different doses administered by intravitreal injection. In 2016, a multicenter study with single dose (10 mg) of CLG561 showed a safety profile on neovascular AMD patients (31 participants), but without complement inhibition [183].

Avacincaptad pegol or ARC1905 or Zimura^^®^^

Avacincaptad pegol (ARC-1905, Zimura^^®^^), a 40 kDa PEG-conjugated aptamer [184], is another inhibitor of C5 complement protein developed by Ophthotech (New York, NY, USA) which can be used in both AMD forms (atrophic and exudative). A phase 1 study (NCT00950638, 2009–2017) [185] proved the safety of avacincaptad pegol up to a dose of 2 mg. Recently (fall 2020), after completion of a phase 2 study (NCT02686658, 2016–2020) [186] evaluating the safety and tolerability of intravitreal injections of avacincaptad, a phase 3 study has been initiated (NCT04435366, 2020–2023) [187] in GA secondary atrophic AMD patients. 

AAV5-VMD2-CR2-fH

AAV5-VMD2-CR2-fH (AAV5-VMD2-mCherry) is an adeno-associated virus gene coding for a C3a inhibitor. In 2018, a study on a mouse model identified a secretion of CR2-fH in RPE cells after injection of AAV5-VMD2-CR2-fH. Moreover, a reduction in the production of C3a associated with CNV was observed. This study shows the potential role of the alternative pathway of complement in the treatment of AMD [188].

AAVCAGsCD59 or HMR59

AAVCAGsCD59 is an adeno-associated viral vector serotype 2 developed by Hemera Biosciences (Newton, MA, USA), which expresses the soluble form of the complement regulatory protein CD59 (sCD59). AAVCAGsCD59 allows normal retinal cells to increase the sCD59 expression, which protects the retinal cells of central vision by inhibiting MAC, complement-mediated cell lysis terminal step. A phase 1 study (NCT03144999, 2017–2021) [189] has been performed to measure intraocular inflammation, evaluate VA, observe GA zone change and its growth rate, the drusen volume in atrophic AMD patient’s eyes and finally the incidence of the conversion of dry AMD to wet AMD. This study showed a well-tolerated profile without serious toxicity and no conversion in neovascular AMD was observed in patients [190].

IONIS-FB-LRX

IONIS-FB-L_RX_ is a ligand-conjugated (LICA) antisense inhibitor of the factor B, a component of the alternative pathway produced in the liver that circulates to the choriocapillaries. A phase 1 study (ACTRN12616000335493, 2020) [191] was performed on 54 volunteers at single and multiple doses of IONIS-FB-L_RX_. A significant reduction in factor B levels in the plasma was observed (56% for 10 mg and 72% for 20 mg at 36 days) without any safety concerns [191]. Two phase 2 studies (NCT03446144, 2018; GOLDEN, NCT03815825, 2019–2022) [192,193] have been initiated to evaluate the safety and efficacy of IONIS-FB-L_RX_ in multiple doses on GA AMD participants. Even if the first was withdrawn rapidly (business issues), the second is currently recruiting. This study will allow the evaluation of the GA area size on 330 patients measured by fundus autofluorescence (FAF).

GT005

GT005 is a recombinant non-replicating adeno-associated viral (AAV) vector encoding a human complement factor I. Three phases 1 or 2 studies (FocuS, NCT03846193, 2019–2026; EXPLORE, NCT04437368, 2020–2024; HORIZON, NCT04566445, 2020–2025) [194,195,196] are ongoing to evaluate the safety, the dose-response, and efficacy of several doses (low, medium, or high) of GT005 administered as a subretinal injection in dry AMD patients.

GEM103

GEM103 is a full-length human recombinant complement factor H protein. A phase 1 study (NCT04246866, 2019–2020) [197] on 12 participants with GA secondary AMD showed a good safety and tolerability profile. Two phase 2 studies (ReGAtta, NCT04643886, NCT04684394, and 2020–2022) [198,199] evaluated GEM103 at multiple doses for geographic and neovascular AMD, respectively. ReGAtta study showed a reduction in biomarkers of complement activation. In addition, supraphysiological levels of factor H have been maintained thanks to GEM103 in both studies while being well-tolerated. 

#### 3.5.2. Other Inflammatory Targets

Sirolimus

Sirolimus (rapamycin, Rapamune^^®^^) is a well-known natural macrocycle exhibiting several therapeutic activities. This inhibitor of the mammalian target of rapamycin (mTOR) [200] was initially used as an immunosuppressor to prevent graft rejections (first agreement by FDA in 1999, given in the case of renal transplantation) [201]. In AMD-treatment, Sirolimus was given in association with Lidocaine (Phase 2, NCT01675947, 2012–2015) [202], but this trial was stopped due to safety concerns. A phase 1/2 study (SIRGA, NCT0071249, 2008–2013) [203], complemented by another phase 1/2 study (SIRGA2, NCT01445548, 2011–2019) [204], evaluated the safety and efficacy of sirolimus formulation; unfortunately, side effects of retinal atrophy and RPE disorders were observed without beneficial effects for patients [205]. Thus, its use remains highly questionable for AMD treatment, and additional studies are required. Recently, in 2021, a study showed a promising subconjunctival delivery profile using sirolimus-loaded PLGA nanoparticles (SIR-PLGA-NP) and chitosan-grafted nanoparticles (SIR-CH-PLGA-NP) in order to achieve a slow release of Sirolimus by ex-vivo scleral penetration [206].

Glatiramer acetate

Glatiramer acetate (Copaxone^^®^^), developed by Teva Pharmaceutical (Tel Aviv, Israel), is a four amino acid (Glu-Ala-Tyr-Lys) polymer mimicking myelin that is currently used in multiple sclerosis (FDA approved in 1996) [207]. Glatiramer acetate suppresses the inflammatory response, as the pro-inflammatory T cells are feigned into Th2 cytokines (anti-inflammatory or regulatory cells). However, the precise mode of action remains elusive [208,209]. In 2007, a phase 2/3 study (NCT00466076, unknown statue) [210] and a phase 1 study (NCT00541333, 2007–2013) [211] assessed glatiramer acetate efficacy and safety for its use as a preventive treatment that blocks the conversion of dry AMD to wet AMD; at the endpoint, both studies highlighted a decrease in drusens’ area [212]. 

Fluocinolone acetonide

Fluocinolone Acetonide (Iluvien^^®^^), is a lipophilic corticosteroïd developed by Alimera Sciences (Alpharetta, GA, USA) [213], FDA approved in 1963 for the treatment of diabetic macular oedema (DME) [214]. In a phase 2 study (NCT00695318, 2008–2015) [215] fluocinolone acetonide (0.2 or 0.5 µg/day) demonstrated a slight improvement in the size of GA after 24 months.

**Table 7 molecules-27-05089-t007:** Drugs targeting inflammation.

Drugs	Structure	Target	Clinic or Research	Formulation	Ref.
Lampalizumab	antigen-binding (Fab) fragment from monoclonal antibody	Factor D	3	Intravitreal injection	NCT01602120 [155]NCT02288559 [152]
Danicopan	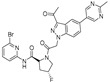	Factor D	2	Oral tablet	NCT05019521 [162]
POT-4	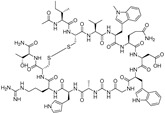	C3	2	Intravitreal injection	NCT01603043 [164]
APL-2	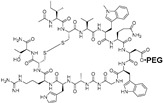	C3	2	Intravitreal injection	NCT02503332 [167]NCT03777332 [169]NCT03525600 [170]
Eculizumab	monoclonal antibody	C5	2	Intravenous	NCT00935883 [175]
Tesidolumab	fully human IgG1, monoclonal antibody	C5	2	Intravitreal injection	NCT01255462 [178]NCT01527500 [179]NCT02515942 [180]
CLG56	Human antibody	Properdin	12	Intravitreal injection	NCT01835015 [182]NCT02515942 [180,183]
Avacincaptad pegol	PEGylated nucleic acid aptamer	C5	123	Intravitreal injection	NCT00950638 [185]NCT02686658 [186]NCT04435366 [187]
AAV5-VMD2-CR2-fH	ocular gene therapy product	Factor H	R	Intravitreal injection	[188]
AAVCAGsCD59	ocular gene therapy product	MAC	1	Intravitreal injection	NCT03144999 [189]
IONIS-FB-L_RX_	ligand-conjugated (LICA) antisense	Factor B	12	Subcutaneously	ACTRN12616000335493 [191]NCT03446144 [192]NCT03815825 [193]
GT005	AAV2	Factor I	1/2	Subretinal injection	NCT03846193 [194]NCT04437368 [195]NCT04566445 [196]
GEM103	Recombinant protein	Factor H	12	Intravitreal injection	NCT04246866 [197]NCT04643886 [198]
Sirolimus	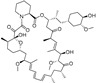	mTOR	2	Intravitreal/Subconjunctival injection	NCT01675947 [202]NCT0071249 [203]NCT01445548 [204]
Glatiramer acetate	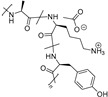	Inflammation	1, 2, 3	Injection	NCT00466076 [210]NCT00541333 [211]
Fluocinolone Acetonide	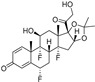	Inflammation	2	Injection	NCT00695318 [215]

### 3.6. RPE and Photoreceptors’ Loss: Stem Cells Curative Strategy

The discovery of stem cells (SC) and their implication in tissues’ regeneration led to the emergence of new therapeutic strategies to restore damaged tissues in numerous diseases (curative strategy vs. symptomatic approach). In the case of the atrophic AMD, three subtypes of stem cells have been studied to regenerate the RPE cells and to produce choriocapillaris and retina maintenance promoting cells [216]: SC may be issued from fibroblasts (iPSC), SC from the central nervous system (HuCNS) and embryonic stem cells (hESC) (Figure 16).

Human-induced pluripotent stem cells (iPSC)

Human-induced pluripotent stem cells (iPSC) (Table 8) are obtained from adult human fibroblasts, resulting from the activation of a set of transcription factors including Oct3/4, Sox2, c-Myc and Klf4 [217]. They share similarities with native RPE cells such as close membrane potential, polarized secretion of VEGF and a comparable gene expression pattern [218,219]. iPSCs can be used to generate pigmented epithelial cells in the outer retina to replace or regenerate the defective ones. Moreover, these cells can produce growth factors and cytokines (brain-derived neurotrophic factor for supportive paracrine effect in the macula) [218]. The iPSC may be sub-retinaly grafted, and a transplantation study in rodent has been carried out on a model of hereditary degenerative retina (RCS rat) [220]. A more recent in vitro study (2016) has been carried out against atrophic AMD to understand its mechanisms. This study relies on models allowing the iPSCs generation from RPE of AMD donors (AMD RPE-iPSC-RPE and AMD Skin-iPSC-RPE) compared to normal RPE-iPSC-RPE. The two AMD-derived models show a different disease phenotype as compared to normal RPE-iPSC-RPE as well as higher ROS formation under oxidative stress. Among others, the SIRT1/PGC-1α pathway seems to play a key role [221], in another trial (NCT02464956, 2015–2019) [222] thereupon. Last, for a better understanding of the cell therapy relevance in AMD, a bank of samples differentiable into ocular cells (transformation of skin, blood into iPSCs) was built (NCT03372746, 2017–2019) [223]. 

Human embryonic stem cells (hESC)

The transplantation of human embryonic stem cells (hESC) (Table 8) has also been tested in atrophic AMD to replace damaged RPEs (MA09-hRPE model, hESC-RPE cell line). Once grafted, the SC proliferated without senescence nor apoptosis [224,225]. A phase 1/2 study (NCT01344993, 2011–2017) [226] demonstrated the survival of the transplant after 12 months, without abnormal proliferation nor side effects for the patients [224,225]. A recent study (NCT02463344, 2015–2020) [227] was led to evaluate the long-term (5 years) safety and tolerability of MA09-hRPE in atrophic AMD patients. Its conclusions have not been published yet. 

CPCB-RPE1 is a subretinal human implant, composed of hESC-RPEs precultured on a biocompatible, mesh-supported submicron parylene C membrane [228]. Their use was assayed in vivo, on Yucatán mini pigs’ eyes, in 2016 and the safety of the subretinal implantation has been proved [228]. A phase 1/2 study (NCT02590692, 2015–2023) [229] is currently ongoing to evaluate the damages of CPCB-RPE1 implant to the RPE/photoreceptor complex in atrophic AMD patients.

Human central nervous system (HuCNS-SC)

The stem cells issued from the human central nervous system (HuCNS-SC) (Table 8) have been used to prevent degeneration of photoreceptors in Royal College of Surgeons rats, animals known for inherited retinal degeneration [230]. The HuCNS-SCs survived over a minimum of eight months after their transplantation into the subretinal space. They minimized long-term vision loss and preserved photoreceptors [231,232]. In humans, the safety profile of HuCNS-SC transplant has been proved by a phase 1/2 study (NCT01632527, 2012–2015) [233]. Another phase 2 study (RADIANT, NCT02467634, 2015–2016) [234] has been implemented by StemCells, Inc to assess potential benefits for AMD patients. The authors concluded that these grafts prevented the aggravation of the symptoms. In some cases, this strategy allowed an improvement in BCVA and in contrast sensitivity (CS), an increase in the central subfield thickness and of the macular volume was also observed. Finally, it showed a decrease in the growth rate of GA in the treated eye compared to the control eye [235]. A study evaluating the long-term safety and possible benefits for patients (NCT02137915, 2014–2016) [236] was terminated prematurely due to financial reasons, despite no safety issues were observed.

**Table 8 molecules-27-05089-t008:** Stem cells therapies against atrophic AMD.

Drugs	Origin	Goal	Clinic or Research	Formulation	Ref.
iPSC	Human fibroblast	RPE cells regeneration	-	Transplantation	NCT02464956 [222]
hESC	Central nervous	Replace RPE cells	1/2	Transplantation	NCT01344993 [226] NCT02463344 [227] NCT02590692 [229]
HuCNS-SC	Fertilization	Photoreceptors regeneration	2	Transplantation	NCT01632527 [233] NCT02467634 [234] NCT02137915 [236]

Conclusion on treatments for atrophic AMD

Despite intensive research towards the treatment of atrophic AMD, no robust therapeutic options (either symptomatic or curative) have emerged yet. To date, only four molecules reached phase 3 clinical trials (ALK-001, trimetazidine, tandospirone, and lampalizumab). The strategy consisting in targeting amyloid β accumulation with monoclonal antibodies (RN6G and GSK933776) showed no beneficial effect. Last, there are no sufficient and convergent results yet, to consider the use of SC in replacing or regenerating damaged cells. Today, only lifestyle improvement is advised to delay disease progression and to prevent its evolution to the exudative form of AMD. It is worthy to note that this evolution is much more detrimental for the life quality of patients. Indeed, the precise mechanism of the atrophic AMD remains poorly understood, which impedes the emergence of new standards for care.

## 4. Ongoing Research and Clinical Trials for the Treatment of Exudative AMD (Wet-AMD)

Exudative AMD has been reported for the first time in the 1850s. This advanced AMD form is more aggressive than the atrophic one, as it may lead to blindness in few months only. The exudative form affects approximately 15% to 20% of AMD patients [47], but is responsible for a large majority of blindness cases. Therefore, finding efficient treatments is an urgent need for public health.

The first treatment was proposed in the 1980s; it concerned laser photocoagulation, which induced severe side-effects [237]. Fortunately, since the beginning of the 21st century, the discovery of new molecular targets and the validation of new drugs are constantly accelerating (Figure 17). From those, only one therapeutic option has been pursued, so far, in wet AMD care: the anti-angiogenic treatments [238,239]. 

Like atrophic treatments, drug formulation is a real challenge and must be anticipated. Currently, all drugs on the market are administered by intravitreal injections, that are invasive and sometimes stressful for patients. To counter this, eye drop formulations are currently being developed for upcoming drugs.

### 4.1. Phototherapies

Laser photocoagulation

Laser photocoagulation, developed in the late 1980s, was the first therapeutic option against exudative AMD [237]. This treatment aims to preserve VA after the appearance of new choroidal vessels in the juxtafoveolar or extrafoveolar regions. It is a non-selective treatment that destroys the new vasculature responsible for both fibroglial scar formation of a and the surrounding neuroretin. Practically, a laser beam is focalized on the RPE; it generates locally an increase in temperature, inducing the destruction of the neo-vessels (Figure 18). Different types of lasers were clinically used, including Argon Blue-green and Argon Green only, Krypton Red and Yellow, and the tuneable Dye Laser [237].

The Macular Photocoagulation Studies (MPS, NCT00000158) [240], a phase 3 clinical trial, initiated in the 1970s with a 5-year follow-up for patients with extrafoveal choroidal neovascularization, demonstrated significant effectiveness of the treatment with the Argon Blue-Green Laser Photocoagulation method. However, a visual decompensation and a loss of VA were observed. In addition, some immediate or delayed side effects, such as scar enlargement or scotoma, were observed [241]. This therapy is no longer used today due to severe neovessels and retina damages.

Photodynamic therapy (PDT)

Photodynamic therapy (PDT) has been originally used to treat cancers [242]. It consists in intravenous administration of photosensitizing agents to patients, followed by their local activation by a laser beam. This results in the formation of ROS inducing localized chemical damages to surrounding biomolecules (proteins, membranes and nucleic acids), leading to cell death. Novartis Pharma S.A.S. extended the PDT scope to the treatment of the choroidal neovascular membrane in the 2000s, and developed verteporfin (Visudyne^^®^^). This photosensitizing agent is able to induce radicals close to the retro-foveal neovessels. Visudyne^^®^^ is administered by intravenous infusion (over 10 min at a dose of 6 mg/m^2^ of body surface area), followed by an activation by 15 min laser beam during. This treatment allows to target a specific vessels’ area without affecting the surrounding tissues [243]. 

Verteporfin is a monoacid derivative of benzoporphyrin (BPD-MA, chlorin type) that exists as a mixture of two regioisomers (BPD-MA_C_ and BPD-MA_D_, Figure 19). The drug lifetime has a half-life of t_1/2_ = 5–6 h in humans and does not present inherent cytotoxicity [243,244,245]. Once activated by a non-thermal red light at 689 nm (diode laser), this drug allows singlet oxygen generation (Figure 20). For many years, this treatment has been studied through various clinical cases [246].

Two studies were performed to evaluate the verteporfin/PDT effects on lesions and VA in patients with exudative AMD: Treatment of Age-related Macular Degeneration with Photodynamic Therapy (TAP) and Verteporfin In Photodynamic Therapy (VIP). Both studies showed that small lesions respond better to this treatment than larger ones. In addition, this treatment allowed a relative maintenance in VA [247]. Visudyne^^®^^ was approved by FDA in 2002 [248] (approved by French HAS in 2000 [249]) for its use against choroidal exudative AMD. An extension was obtained from French HAS in 2002 for occult retrofoveolar choroidal neovascularization [250]. 

Phototherapy is not used anymore as a treatment in AMD patients, due to the significant higher efficacy of anti-VEGF drugs. 

### 4.2. Anti-VEGF Drugs

Angiogenesis is one of the major key markers of the progression to wet AMD. The breakthrough discovery of anti-VEGF drugs brought real improvement for patients with both cancer and angiogenic eye diseases [251]. Thus, new therapeutic options for wet AMD emerged in the 1990s. A particular emphasis has been dedicated to research on the Vascular Endothelial Growth Factor A (VEGF-A), which is one of the major key players in pathological angiogenesis [238,239]. However, different VEGF-A isoforms exist: some of them exerting pro-angiogenic effects (VEGF-A_206_, VEGF-A_189_, VEGF-A_165,_ and VEGF-A_121_), while others are considered as less angiogenic or anti-angiogenic (VEGF-A_186b_, BEGF-A_165b_, and VEGF-A_121b_). In homology with cancers and retinopathies, the most active isoform in AMD is the overexpressed pro-angiogenic VEGF-A_165_. The main drawback of anti-VEGF drugs lies in the fact that these therapeutic agents remain symptomatic, not curative, and they do not induce regeneration in VA. Secondary limitations may occur due to long term intravitreal injections treatments such as the occurrence of retina atrophy [252]. Despite these disadvantages, these drugs bring indisputable benefits for responder patients and remain the standard wet AMD care.

#### 4.2.1. Marketed Drugs

Currently, five molecules have been approved for the symptomatic treatment of wet AMD, all of them being large size biologics.

Pegaptanib sodium (Macugen^^®^^)

Pegaptanib sodium (Macugen^^®^^, developed by Pfizer, New York, NY, USA) is a short (28-base) RNA oligonucleotide PEGylated aptamer, which specifically targets VEGF-A_165_ [253] (IC_50_ = 750–1400 pM) [254]. The key study VISION (VEGF Inhibition Study in Ocular Neovascularization) explored the intravitreal use of pegaptanib sodium (0.3; 1; or 3 mg every six weeks) for VEGF inhibition in subfoveal CNV [255]. These treatments resulted in patients’ VA improvement. Thus, the drug was approved by the FDA in 2004 as 0.3 mg injectable solution [256] and was marketed in France in 2006 to treat wet AMD. Nevertheless, this drug was discontinued due to its lack of efficacy compared to other anti-VEGF.

Ranibizumab (Lucentis^^®^^)

Ranibizumab (Lucentis^^®^^), also called rhuFab V2, was developed by Novartis Pharma (Basel, Switzerland). This recombinant humanized monoclonal antibody of 48 kDa targets all the VEGF-A isoforms (IC_50_ ranging between 88 pM and 1140 pM) [254]. The most relevant study is a phase 3 clinical trial (MARINA, NCT00056836, 2003–2014; NCT01442064, 2011–2012; NCT00379795, 2006–2017) exploring its use by intravitreal injection (0.5 mg monthly) [257,258,259]. This 2-years study demonstrated (i) a prevention of vision loss, (ii) an average VA improvement, and (iii) a reduction in angiographic lesions. Minor side effects have been reported (intraocular inflammation for 8–14.6%, phakic eyes for 5.1–7.2% patients, and endophthalmitis for 1%, were among the most relevant) [260].

The ANCHOR study (NCT00061594, 2003–2014) [261] compared the use of ranibizumab with verteporfin use for photodynamic therapy (PDT) with verteporfin. This study undoubtedly established that ranibizumab improves VA of 1 year in average, and exhibits superior beneficial effects than PDT. Few low ocular adverse effects were observed (intraocular inflammation for 8–9.3%, rhegmatogenous retinal detachment for 0.7%, endophthalmitis for 0.7%) [262]. The long-term (7–8 years) prognosis of these treatments has been summarized by the SEVEN-UP study, which concluded that one-third of the patients demonstrated good visual outcomes, whereas another third had poor outcomes [263]. Afterwards, different drug administration protocols have been studied for 0.5 mg: HARBOR (13.5 mean number injections showing an improvement in BCVA) [264] and TREX-AMD (13 and 10.1 mean number injections showing visual gain) [265]. Moreover, a delivery system has been designed based on magnetic nanoparticles loaded with PEG-PLGA copolymer matrix. This study showed an inhibition of tube formation highlighting the promising profile of nanoparticles for neovascular AMD [266]. A port delivery system (PDS) has been studied for ranibizumab, based on eye implant allowing the continuously delivery over several months to reduce the number of intravitreal injections [267].

Ranibizumab was FDA-approved in 2006 and marketed in 2007 in France for the treatment of wet AMD under the brand name Lucentis^^®^^ (injectable solutions of 0.5 mg/month). Importantly, this drug is also used to treat neovascularization secondary choroid for high myopia (MF), diabetic macular edema (MDG), macular edema secondary to retinal vein branch occlusion (OBVR), or central retinal vein (OVCR) [268].

Aflibercept (Eylea^^®^^)

Aflibercept was developed by Bayer HealthCare (Leverkusen, Germany) under the brand name Eylea^^®^^. This is a dimeric recombinant fusion glycoprotein, composed by fragments of the VEGF receptors 1 and 2 extracellular domains with an IC_50_ of 16–90 pM [254]. These domains are fused thanks to the Fc fusion protein (115 kDa) region of human immunoglobulin gamma 1 (IgG1) [269]. 

Phase 3 studies comparing ranibizumab and aflibercept (VIEW1/2, NCT00509795, 2007–2012; NCT00637377, 2008–2014) [270,271] proved that aflibercept intravitreal doses (0.5 mg to 2 mg) administered every 2 months (after 3 initial monthly doses) show similar efficacy and safety than monthly ranibizumab (0.5 mg) treatment. This lower injection frequency brought indisputable improvements for the patients (better tolerance after successive injections, less side effects in the eyes) [269].

Aflibercept was FDA-approved in 2011 and marketed in France in 2012 to treat wet AMD (injectable solution of 2 mg/month). Interestingly, aflibercept is also used in oncology (Zaltrap^^®^^), to treat adults with metastatic colorectal cancer (in combination with fluoropyrimidine, MA in 2005) or metastatic breast cancer (in combination with paclitaxel, MA in 2007) [250]. Moreover, it has been used as treatment for macular edema following retinal vein occlusion (RVO), macular edema (DME), and diabetic retinopathy (DR) in patients with DME [272].

Bevacizumab (Avastin^^®^^)

Bevacizumab or rhumAb was developed by Roche (Basel, Switzerland). This is a recombinant humanized full-length 149 kDa monoclonal antibody directed toward all VEGF-A isoform [273] with an IC_50_ of 500–1476 pM [254]. This mAb has been marketed in 2004 (Avastin^^®^^) for the treatment of metastatic colorectal cancer and other neoplastic diseases [274].

Five years of study of CATT (comparison of age-related macular degeneration treatments trials), comparing the use of ranibizumab vs. bevacizumab, showed that these drugs have similar effects on VA [275,276]. A phase 3 study (GEFAL, NCT01170767, 2010–2019) [277,278] compared ranibizumab and bevacizumab to evaluate the efficacy in clinical terms on the VA in exudative AMD patients [277,278]. These results led to its clinical use for the treatment of exudative AMD, either off-label or via a temporary authorization in France (2015) because of market competition concerns with Lucentis^^®^^. It is available as intravitreal administration like other anti-VEGFs, contrary to its intravenous use for colorectal metastatic or angiogenic cancers [279]. 

Brolucizumab (ESBA 1008, RTH 258, Beovu^^®^^)

Brolucizumab (Beovu^^®^^) was developed by Novartis (Basel, Switzerland). This is a relatively low molecular weight (26 kDa) humanized single-chain antibody fragment, that inhibits all isoforms of VEGF-A (IC_50_ = 0.49 nM) [280,281].

Recently, four clinical trials (one phase 2 trial in 2018: NCT02507388 and three phases 3 trials in 2019 and 2020: NCT03930641, NCT03954626, NCT03386474) [282,283,284,285] were completed and showed the safety of 6 mg of brolucizumab in exudative AMD patients. Moreover, the two phases 3 trials HAWK (NCT02307682, 2019) [286] and HARRIER (NCT02434328, 2019) [287] compared the effects of brolucizumab to aflibercept [288]. These studies differ in the administered intravitreal doses: two doses of brolucizumab (3 mg and 6 mg) and a dose of 2 mg aflibercept for HAWK; 6 mg of brolucizumab vs. a dose of 2 mg of aflibercept for HARRIER. Overall, these studies demonstrated that at 6 mg brolucizumab is an interesting anti-VEGF drug with efficacy, safety and solubility that can be compared to the 3 other anti-VEGF that are currently in use for wet AMD treatment (ranibizumab, aflibercept, and bevacizumab). Moreover, HAWK and HARRIER show that brolucizumab may be a more durable agent compared to other anti-VEGF, which allows high molar doses, and thus fewer injections for patients. Several phase 3 or 4 clinical trials have been recruited in 2021 to assess the effects of brolucizumab on patients with exudative AMD.

Brolucizumab was recently approved in US (FDA) and Europe for the treatment of exudative AMD (2019); however, without possibility of reimbursement. For this reason, combined with inflammatory side-effects and retinal vascularity sometimes occlusive, its clinical use remains very limited [289].

To summarize, among the five anti-VEGF agents which have been approved for the treatment of wet-AMD from 2006 to 2019, four are marketed (Table 9). All these drugs are administered by intravitreal injection, which allows a focused action. However, these treatments might be traumatic for some patients [290] and are not curative, since they can only slow down or pause the progression of the disease. Last, ranibizumab and bevacizumab have similar effects on VA after 1/2 years and similar side effects [275,291]. Injection price discrepancies are nevertheless of note, bevacizumab being significantly more economical.

#### 4.2.2. Future Trends

In addition to the currently marketed anti-VEGF treatments, several drugs were clinically tested and are depicted below.

Proteins drugs

Conbercept (Lumitin^^®^^)

Conbercept (Lumitin^^®^^, also called KH-902) is developed by Chengdu Kanghong Biotech Co., Ltd. (Chengdu, China). This is a recombinant fusion protein similar to aflibercept, with a larger VEGF2 domain 4 [292]. It is composed of the second Ig domain of VEGFR-1, the third and fourth Ig domains of VEGFR-2, and the constant region (Fc) of human IgG1. Like aflibercept, conbercept blocks all isoforms of VEGF A, VEGF B, and PIGF but with a higher affinity than bevacizumab and ranibizumab (IC_50_ = 7–15 pM) [293]. Moreover, this mAbs exerts doubly superior bioactivity for VEGF inhibition compared to ranibizumab [294]. 

After in vivo studies (rats and mice) [293], conbercept has been evaluated in several clinical studies in phase 1 (HOPE, NCT01242254, 2010–2014) [292,295], 2 (AURORA, NCT01157715, 2010–2014) [296,297], 3 (PHOENIX, NCT01436864, 2011–2014) [298], and 4 (RELIANCE, NCT02577107, 2015–2017) [299]. However, the phase 3 dose-ranging trials, PANDA (NCT03577899, NCT03630952, 2018–2021) [300,301] were recently stopped (2021) since the primary endpoints have not been reached (only 40% of the recorded cases followed the dosing regimen).

This drug was approved by the China Food and Drug Administration in 2013 but not marketed in the United States nor in France because worldwide clinical studies have not been provided yet [273,294,302].

Abicipar pegol

Abicipar pegol (or AGN-150998) is a recombinant protein of the DARPin family (Designed Ankyrin Repeat Protein) developed by Allergan (Dublin, Ireland). It is a small 34 kDa protein formed by a repeated ankyrin structures. This recombinant protein has a greater affinity, for all the VEGF-A isoforms, than ranibizumab (Kd = 2 pM vs. 46 pM) [303]. In 2018, Allergan demonstrated that abicipar pegol has also a longer duration of action than ranibizumab and bevacizumab [304]. 

Phase 1 studies (NCT02859766, 2016–2018; PINE, NCT03335852, 2017–2018) [305,306] evaluated the safety and pharmacokinetics of abicipar pegol in exudative AMD, the results have not been released yet. Phase 2 studies (CYPRESS, NCT02181517, 2014–2016; BAMBOO, NCT02181504, 2014–2017) [307,308] has been implemented in Japan and in the US to compare the effects of three monthly intravitreal injections of abicipar (1 or 2 mg) or five monthly intravitreal injections of ranibizumab (0.5 mg). This study established that abicipar pegol not only induces an improvement in VA but also a decrease in the thickness of the retina [303]. In addition, a phase 2 stage 3 study (REACH, NCT01397409, 2011–2019) [309], demonstrated that abicipar pegol reduces the number of the injection compared to ranibizumab treatment (3 against 5), while improving the BCVA and the thickness of the central retina (CRT). In addition, no serious side effects have been reported [310]. Finally, two phases 3 studies conducted with a larger dose of abicipar pegol (2 mg) at different injection times, showed an enhancement and stability of VA with a lower administration dose of abicipar pegol compared to ranibizumab (CEDER, NCT02462928, 2015–2019; SEQUOIA, NCT02462486, 2015–2019) [311,312]. Even if the incidence of intraocular inflammation (IOI) is higher with abicipar pegol, this drug allows quarterly injections while ranibizumab is administered monthly [313]. Moreover, a 28-week phase 2 study (MAPLE, NCT03539549, 2018–2020) [314] has been recently completed, highlighting a lower IOI incidence than phase 3 in abicipar-treated patients. Nevertheless, these studies were stopped by Allergan due to a number of intraocular inflammatory adverse events.

OPT-302

All the above-mentioned drugs target mainly VEGF-A, however AMD is also characterized by abnormally elevated levels of other endothelial growth factors, such as VEGF-C and VEGF-D (and their corresponding receptors VEGFR-2 and VEGFR-3), that are also key players of hyper neo-vascularization. Moreover, VEGF-C is reported as a potent inducer of vascular permeability or leakage. Thus, blocking simultaneously VEGF-A, C and D could stop the angiogenesis and the vascular leakage occurring in wet AMD. In line with these considerations, OPT-302 has been developed by Opthea (Melbourne, Australia). This is a soluble form of VEGFR-3 (the transmembrane receptor for VEGF-C and VEGF-D), including the 1–3 extracellular domains of VEGFR 3 and the Fc fragment of human IgG1. OPT-302 is designed to bind and consequently neutralize the activity of VEGF-C and D. The safety, pharmacokinetics, and pharmacodynamics of OPT-302 have been evaluated via a phase 1 trial (NCT02543229, 2015–2017) [315,316] with an intravitreal injection given at a 2 mg dose, alone or in combination with ranibizumab (0.5 mg). Then, a phase 2 study (NCT03345082, 2017–2020) [317] was carried out to compare two doses of OPT-302 (2 mg and 0.5 mg) in combination with ranibizumab (0.5 mg). This trial revealed that the combined therapy induces a better VA, a decrease in macular thickness with OP-302-ranibizumab association compared to the use of ranibizumab alone. Moreover, this strategy allows a longer delay between two successive injections (every 4 weeks).

Small-sized molecules

Another anti-angiogenic strategy consists in targeting growth factor receptors with small-sized molecules. Indeed, these receptors exert intracellular tyrosine kinase inhibiting activities on several substrates and are involved in numerous signaling pathways. This results in proliferation, cell growth, apoptosis, angiogenesis, and cellular motricity. These growth factors receptors have been extensively studied for more than two decades as relevant targets in oncology for the development of “targeted therapies”. 

Sorafenib

Sorafenib (BAY 43-9006, Nexavar^^®^^), is a polyspecific kinases inhibitor of Raf, PDGF, VEGF receptors, and c-Kit, developed by Bayer (Leverkusen, Germany) with low nanomolar IC_50_ (26, 90, and 20 nM for VEGFR-1, -2, and -3, respectively) [318]. This molecule has been initially approved by the FDA in 2005 for its use against hepatocellular carcinoma, renal cell carcinoma, and thyroid carcinoma [319]. In France, this drug has been marketed for the treatment of hepatocellular carcinoma and renal cell carcinoma in 2006 [320]. 

In ocular diseases, an in vitro study was performed to determine the effects of a simultaneous blockade of VEGF, PlGF, and PDGF. By treating RPE cells with sorafenib and exposing them to white light, the levels in these abovementioned growth factors are reduced. In contrast, with the same conditions, the absence of sorafenib led to an increase in VEGF, PIGF, and PDGF. This result highlights the potential benefits of sorafenib in AMD [321]. 

Moreover, some studies have been performed on isolated cases, consisting of the use of sorafenib in association with other anti-VEGF; a study was implemented in 2008 to monitor the effects of a treatment combining sorafenib (200 mg, 3 times a week, for 5 weeks) and ranibizumab. This study showed that, in combination or alone, sorafenib allows VA improvement and a resolution of the intraretinal fluid without serious adverse effects. Another study on a single case was performed in 2008 to evaluate the effect of a bevacizumab injection (1.25 mg twice) in combination with oral sorafenib (400 mg twice daily). A VA improvement and a decrease in the retina thickness were observed. A longer-term study would be required for sorafenib use [322]. However, to unambiguously validate these results, another clinical study would be necessary with a larger recruitment of AMD patients [323].

Pazopanib

Pazopanib (Votrient^^®^^) is another small-sized inhibitor that targets VEGFR-1, -2, and -3 (IC_50_ = 7, 15, 2 nM), platelet-derived growth factor (PDGFR-α, -β; IC_50_ = 73, 215 nM) [324], and stem cell growth factor (c-Kit). This molecule has been developed by GlaxoSmithKline (GSK, Brentford, UK) and has been approved by the FDA in 2009 for the treatment of the renal cell carcinoma and of the soft tissue sarcoma (in combination with chemotherapy) [325]. 

GSK led several studies on the oral administration of pazopanib to evaluate its safety, tolerability, efficacy, absorption, pharmacodynamics or pharmacokinetics for exudative AMD: phase 1 studies (NCT00463320, 2007–2012; NCT00659555, 2008–2017; NCT01154062, 2010–2017; NCT01051700, 2010–2017; NCT01072214, 2010–2017) [326,327,328,329,330] and phase 2 studies (NCT00612456, 2008–2017; NCT00733304, 2008–2017; NCT01134055, 2010–2018) [331,332,333], (NCT01362348, 2011–2017) [334]. These trials showed that pazopanib is well tolerated with improvement in BCVA, central retinal lesion thickness, and central retinal thickness [335]. However, a phase 2 study [334] was stopped because of a lack of efficacy [336]. Finally, another phase 2 study [333] compared the effects of pazopanib given in eye drops versus intravitreal injections of ranibizumab. This study reveals that pazopanib drops are well tolerated; however, after using pazopanib drops for one year, no greater therapeutic improvement has been observed and the number of intravitreal injections of ranibizumab could not be reduced [337]. 

Axitinib

Axitinib (Inlyta^^®^^) is a small-sized inhibitor of VEGFR, PDGFR and fibroblast growth factor receptor (FGFR), colony-stimulating factor receptor developed by Pfizer (New York, NY, USA). This molecule demonstrates a higher inhibiting potency of kinases compared to some reference compounds such as pazopanib and sorafenib. Indeed, axitinib inhibits VEGFR-1, -2, and -3 with IC_50_ ranging between 0.1 nM to 0.3 nM, and inhibits PDFG-R, FGFR with IC_50_ ranging between 1.6 nM to more than 1000 nM for PDGF, FGFR, colony-stimulating factor [338]. The FDA approved its use in 2012 to treat renal cell carcinoma [339]. In vitro and in vivo studies on a mouse model of AMD demonstrated that oral administration of axitinib enables 70.1% inhibition of choroidal neovascularization (CNV) lesions and significant regression of established CNV [340,341]. Axitinib could be used in combination with other anti-VEGFs in the treatment of exudative AMD but more results are needed to determine its safety, the dose or long-term outcomes of such treatment [341]. A recent phase 1/2 was initiated in 2020 (OASIS, NCT04626128, 2020–2022; NCT05131646, 2021–2022) [342,343] to evaluate the safety and tolerability of a suprachoroidally formulation of axitinib in AMD patients, whom have been treated with anti-VEGF drug. Moreover, in order to evaluate another formulation with sustained release, Ocular Therapeutix has been recently started a phase 1 study based of a dried polyethylene glycol-based hydrogel fiber containing axitinib: OTX-TKI (NCT04989699, 2021–2022) [344].

Acrizanib

Acrizanib (LHA510) is a small-molecule VEGFR-2 inhibitor (IC_50_ = 17.4 nM, BaF3-VEGFR-2) which inhibits CNV in mouse (99%) and rat (94%) models [345]. Moreover, an exposition over one eye of rabbits for 7 days of acrizanib (2% suspension, 30 μL) showed an encouraging ocular pharmacokinetics profile [345]. A phase 1 study (NCT02076919, 2014) [346] and a phase 2 study (NCT02355028, 2015–2016) [347] have evaluated the topical administration compared to invitreal injection. The results were not satisfying with adverse events such as primarily corneal haze and/or edema [348].

Regorafenib

Regorafenib (Stivarga^^®^^) is a small-sized kinases inhibitor developed by Bayer (Leverkusen, Germany), which remarkably inhibits VEGFR-1, -2, -3 (IC_50_ = 13, 4, 46 nM) and PDGFR (IC_50_ = 22 nM) [349]. This drug has been approved by the FDA for the treatment of metastatic colorectal cancer (CRC), of gastrointestinal stromal tumor (GIST) in 2012, and of hepatocellular carcinoma (HCC) in 2017 [350]. In the case of AMD, a phase 2 clinical trial (DREAM, NCT02222207, 2014–2016) [351] evaluated its use as eye drops. However, this study was prematurely terminated due to a lack of efficacy compared to current treatments, potentially due to an unsatisfactory diffusion of the molecule in the posterior segment of the eye [352].

SH-11037

SH-11037 is a synthetic derivative of the antiangiogenic homoisoflavonoid cremastranone, which inhibits ocular angiogenesis in zebrafish larvae at 10 μM (40% reduction in hyaloid vessel) [353]. In the mouse laser-induced-CNV model, the CNV lesion has been reduced thanks to SH-11037 at 1 and 10 μM (42% and 55% of reduction respectively) with reduced leakiness of CNV lesions. Moreover, SH-11037 presents a safe and effective profile without toxicity issues (up to 100 μM, 14 days) [353]. In 2018, a study showed an epoxy fatty acid metabolism enzyme as a new target for neovascular AMD: the soluble epoxide hydrolase (sHE). SH-11037 binds almost the entire active site of sHE in hydrolase domain and inhibits its activity in vitro (IC_50_ > 10 μM) and in vivo this enzyme in CNV mouse model (at 10 μM). sHE is expressed in human neovascular AMD, in consequence, it may constitute a relevant target to counter the disease [354].

PAN-90806

PAN-90806, is a VEGF-R2 inhibitor (IC_50_ = 1.27 nM) developed by PanOptica, Inc. (Mount Arlington, NJ, USA) for the treatment of exudative AMD and other neovascular eye diseases, exerting an antiangiogenic effect [355]. PAN-90806 is an interesting molecule due to its topical administration by eye drops [356]. A phase 1 study (NCT02022540, 2013–2016) [357] has been completed in 2016 to evaluate the safety and tolerability of PAN-90806, as monotherapy or in association with ranibizumab. It showed a biological positive response for 45–50% of patients. After this phase 1 outcomes, a recent phase 1/2 (NCT03479372, 2018–2019) [358] has been launched to determine the optimal dose of PAN-90806 to be orally administered. Thus, for a once-daily administration at 2, 6, and 10 mg/mL concentrations, PAN-90806 is safe and well-tolerated.

Vorolanib

Vorolanib (X82 or CM-082) is a kinase inhibitor derived from sunitinib, with anti-VEGFR (IC_50_ = 0.052 μM) and anti-PDGFR (IC_50_ = 0.26 μM) activities [359,360]. Tyrogenex, Inc. initiated two trials: a phase 1/2 study (NCT01674569, 2012–2018) [361] to evaluate the use of vorolanib in combination therapy with ranibizumab, and a phase 2 study (NCT02348359, 2015–2018) [217] in which vorolanib was given in combination with one of the three anti-VEGF approved in AMD treatment (ranibizumab, aflibercept and bevacizumab). For the patients who completed these studies (25 participants, 71%), the authors reported a maintenance or an improvement in VA, as well as a reduction in the average thickness of the central subfield (SD), except in one case. Unfortunately, these positive results are counterbalanced by undesirable side effects including diarrhea, nausea, fatigue, and transaminase increase, which led to the discontinuation of the treatment for some patients (17%). Thus, additional studies are needed to evaluate the balance between safety and efficacy [359]. In 2021, EyePoint Pharmaceuticals started a phase 1 study (NCT04747197, 2021–2022) [362] to evaluate EYP-1901 solution (440, 2060 and 3090 μg) consisting in vorolanib delivery in a Durasert bioerodible TKI.

Moreover, another drug was developed by Graybug Vision (Baltimore, MD, USA): GB-102, a poly lactic-co-glycolic acid microparticles containing sunitinib. Two studies in phases 1 and 2 respectively (NCT03249740, 2017–2019; ALTISSIMO, NCT03953079, 2019–2022) [363,364] showed a potent and 6-month durable safety profile. In the ALTISSIMO study, GB-102 was injected by intravitreal injection every 6 months (1 and 2 mg) in comparison with aflibercept (2 mg). Despite the absence of side effects (inflammation, intraocular pressure), a diminution of VA (decrease of 9 ETDRS letters in the assay) upon treatment with GB-102. A 6-month prolongation is ongoing [365]. 

Lenvatinib

Lenvatinib (E7080 or Lenvima^^®^^) is developed by Eisai Inc. (Tokyo, Japan) and was approved by FDA in 2015 for differentiated thyroid cancer (DTC), renal cell carcinoma (RCC), and hepatocellular carcinoma (HCC) [366]. This is a kinase inhibitor [367] that targets VEGFR-1, -2, and -3, fibroblast growth factor receptor (FGFR1), and platelet-derived growth factor receptor (PDGFR) with an IC_50_ in the 20 μM range [368]. This drug is interesting due to its ability to cross the blood–retina barrier. Thanks to an in vivo zebrafish model, it has been reported that lenvatinib (5 μM or 10 μM) inhibits angiogenesis without toxicity after 48 h [369]. Finally, lenvatinib also stops CNV in a neovascular AMD mouse model (10 mg/day), confirming its interest in exudative AMD treatment.

Brivanib

Brivanib alaninate (BMS-582664) is an orally available inhibitor of FGFR and VEGFR (IC_50_ = 34 nM) [370], with a marked specificity against VEGFR2 and FGFR1 [371], developed initially by Bristol–Myers Squibb to treat hepatocellular carcinoma. Studies on mouse and zebrafish models have shown that brivanib is also an inhibitor of proliferation, migration, tubule formation of choroidal microvascular endothelial cells and angiogenesis as observed with lenvatinib [372]. These recent results require more in-depth studies such as the analysis of brivanib metabolism in the mouse body, the mechanisms of CNV reduction by brivanib or the monitoring of its effects against others VEGFR/FGFR.

Squalamine lactate (Evizon, OHR-102, MSI-1256F)

Squalamine, developed by Genaera Corporation (Plymouth Meeting, PA USA) and Ohr Pharmaceutical (New York, NY, USA), is a small natural steroid with a steroid-polyamine motive [373], isolated from the liver of sharks (genus Squalus). This drug binds calmodulin once absorbed in the cell, which blocks VEGF signal transduction (antiangiogenic action). This aminosterol is positively charged on the amine functions, which allows electrostatic binding to the negatively charged cell membrane. As a result, the cell migration process is blocked and consequently angiogenesis [374].

Three phase 2 studies (NCT00089830, 2004–2007; NCT00333476, 2006–2007; NCT00094120, 2004–2008) [375,376,377] and a phase 3 study (NCT00139282, 2005–2007) [378] were completed. These studies evaluated the safety and the efficacy of squalamine lactate (40 mg/infusion/week for 4 weeks), but they were stopped because no vision improvement has been reported as quickly as with other treatments.

A phase 2 study (IMPACT, NCT01678963, 2012–2015) [379] compared the use of squalamine (0.2% ophthalmic solution) given in combination with ranibizumab vs. the use of ranibizumab alone. The combination therapy allowed a VA improvement for 42% of patients, compared to 28% for patients receiving ranibizumab alone. This improvement was dependent on the size and type of lesions [380]. A phase 3 study (MAKO, NCT02727881, 2016–2017) [381] was initiated with a 0.2% ophthalmic solution of squalamine injected twice a day, in combination with a monthly injection of ranibizumab. This study showed a lack of effectiveness of VA gain after nine months [382].

siRNA target

Another antiangiogenic strategy consists in the blockade of VEGF-A production thanks to small interfering RNAs (siRNA), which may induce the inhibition of genes coding for this growth factor. However, these agents block the synthesis of new VEGF but do not eliminate pre-existing VEGF, rendering the combinational use of a conventional anti-VEGF agent necessary [383].

Bevasiranib

Bevasiranib, developed by OPKO Health, Inc. (Miami, FL, USA), is a siRNA-based anti-angiogenic agent targeting VEGF-A [383]. Bevasiranib showed interesting safety and efficacy profiles with intravitreal injections in pre-clinical mouse models and clinical studies (phase 1 and 2, without the literature data). Thus, the first siRNA tested in phase 3, was assayed against exudative AMD in association with ranibizumab in a phase 3 trial (COBALT, NCT00499590, 2017–2019) [384]; nevertheless, this clinical trial was interrupted due to a lack of efficacy for this combination therapy.

siRNA-027 or AGN211745

siRNA-027 or AGN211745 has been developed by Allergan (Dublin, Ireland) to suppress CNV and retinal neovascularization by reducing the levels of VEGF-R1-mRNA (IC_50_ = 50 pM) [385].

Two clinical trials have been completed in 2008 (phase 1/2, NCT00363714) [386] and 2015 (phase 2, NCT00395057) [387]. In the first study, single intravitreal injections were given to patients while the second trial studied its co-administration in combination with ranibizumab. These studies were terminated prematurely due to a lack of efficacy in phase 2 without safety issues [388].

Gene therapiesOXB-201 or RetinoStat^®^

OXB-201 (RetinoStat^^®^^), developed by Oxford BioMedica (Oxford, UK), corresponds to a lentiviral vector. The infection by a lentivirus (equine infectious anemia virus, EIAV) leads to the expression of two 20-kDa proteins, whose C-terminal fragments are derived from type XVIII collagen (endostatin) for the first and from plasminogen for the second (angiostatin). These two proteins exert an anti-angiogenic action by blocking endothelial cell proliferation and migration. Moreover, endostatin and angiostatin induce endothelial cell apoptosis and cell cycle arrest. No dose-limiting effects were observed, probably due to the topic subretinal administration. Endostatin and angiostatin expressions in the eye may reduce fluorescein angiographic leakage. However, in the context of exudative AMD, subretinal and/or intraretinal fluid was not removed reliably [389]. This treatment is also used in cancers, diabetic retinopathy in addition to macular degeneration.

A phase 1 study (GEM, NCT01301443) [390], completed in 2017, aimed at the identification of the maximally tolerated dose (MTD) of RetinoStat^^®^^ when administrated by single subretinal injection. This study showed that the EIAV subretinal injections were safe and well-tolerated, and they were reproducible, and sustained transgene expression. Moreover, endostatin and angiostatin allowed the fluorescein angiographic leakage reduction but, subretinal/intraretinal fluid was not eliminated in patients with exudative AMD [389].

CRISPR-Cas9 ribonucleoproteins (RNPs)

Cas9 (clustered, regularly interspaced, short palindromic repeat)/Cas (CRISPR-associated)) ribonucleoproteins (RNPs) can be delivered to human stem and primary cells, as well as to mice to modify target genes [391,392]. RNP Cas9 was administered in an in vivo study (2017) by subretinal injection into RPE cells in adult mice for a potential local treatment of AMD [393]. The first results showed that CNV is reduced in treated animals and that in RPE, the target site, underwent mutagenesis. However, other studies on other species are required to verify the initial theories supported by this study.

rAAV.sFlt-1

rAAV.sFlt-1 is a recombinant adeno-associated virus (rAAV2) vector encoded with a naturally occurring VEGF inhibitor known as soluble VEGFR1 receptor or sFLT-1. A phase 1/2 study (NCT01494805) [394] was completed in 2017 by Lions Eye Institute, Perth, Western Australia and Adverum Biotechnologies, Inc., in which rAAV.sFlt-1 was administrated by subretinal injection at low or high dose (1 × 10^10^ or 1 × 10^11^ vector genomes of rAAV.sFlt-1). This study showed a good tolerance for this treatment and demonstrated a favorable safety profile [395].

AAV.sFLT-01

AAV.sFLT-01 is an adeno-associated viral (AAV) vector similar to rAAV.sFlt-1, with parts of sFLT1 grafted onto an Fc fragment. A phase 1 study (NCT01024998) [396], was completed in 2018 by Genzyme, in which AAV.sFLT-01 was administrated by intravitreal injection. The treatment seemed safe and well-tolerated at all tested doses (2 × 10^8^, 2 × 10^9^, 6 × 10^9^ and 2 × 10^10^ vector genomes AAV2-sFLT01) [397]. However, expression variability and anti-permeability activity were reported, and more detailed studies must be developed to identify their causes. In addition, the potential effect of the basic anti-AAV2 serum antibodies should be studied.

ADVM-022

ADVM-022 (AAV.7m8-aflibercept) is a novel recombinant adeno-associated virus (AAV) developed explicitly for exudative AMD. This gene therapy is based on the utilization of AAV2.7m8 capsid and intravitreal injections allowing the retinal transduction to deliver aflibercept [398]. ADVM-022 is being currently studied through a phase 1 clinical study (OPTIC, NCT03748784, 2018–2022) [399] developed by Adverum Biotechnologies (Redwood City, CA, USA). Two doses are administrated to compare central vision (BCVA) improvement or maintenance, and the injection frequencies. However, severe toxicity side effects have been recently noted with acute retinal necrosis and inflammation [400].

RGX-314

RGX-314 is a recombinant adeno-associated virus (AAV8) comprising a gene encoding for a monoclonal antibody fragment similar to ranibizumab. Regenxbio Inc. explored this AAV8 anti-VEGF in clinical trials with different administration modes: a suprachoroidally phase 1 (AAVIATE, NCT04514653, 2020–2023) [401], and a subretinally phase 2/3 (ATMOSPHERE, NCT04704921, 2021–2024) [402]. These two studies will evaluate the efficacy and safety of RGX-314 in patients with exudative AMD.

### 4.3. Anti-PDGFs

Several studies pointed out that PDGF (platelet-derived growth factor) is upregulated in response to anti-VEGF therapy. This up-regulation induces the formation of a protective layer of pericytes to cover the neovascular complex, promoting in turn a resistance to anti-VEGF therapies [403]. Hence, it may be important to consider pretreatment with an anti-PDGF to avoid anti-VEGF resistance to these therapies. This strategy generally improves visual outcomes [404].

Pegleranib or E10030 (Fovista^®^)

Pegleranib is a 29-nucleotides modified DNA PEGylated aptamer of 40 kDa substituted by 2′-fluoro- and 2′-O-methyl- [184], developed by Ophthotech (New York, NY, USA). This PDGF antagonist blocks the interaction with its receptor, the PDGFR-β [405]. Several trials have been performed to study its use in combination with ranibizumab. The phase 1 study (NCT00569140, 2007–2010) [406] and the phase 2 study (NCT01089517, 2010–2017) [407] demonstrated an improvement in BCVA. However, the phase 3 clinical studies (NCT01940900, 2013–2018; NCT01944839, 2013–2018; NCT01940887, 2013–2020) [184,408,409,410] did not conclude to benefits in VA after 12 months compared to the use of aflibercept or bevacizumab alone.

Rinucumab or REGN2176-3

Rinucumab is a monoclonal antibody developed by Regeneron (Tarrytown, NY, USA) directed towards PDGF. A phase 1 study (NCT02061865, 2014–2015) [411] demonstrated a VA maintenance without serious adverse effects. Then in a subsequent phase 2 study, rinucumab was assayed in combination with aflibercept (CAPELLA, NCT02418754, 2015–2017) [408]; however, this trial showed no benefit of the combination compared to a monotherapy of aflibercept.

### 4.4. Angiopoietin 2 Inhibitors

Angiopoietin-2 (Ang-2) is a cytokine involved in angiogenesis and inflammatory processes, whose Tie-2 receptor- is expressed by endothelial cells and fibroblasts [409]. In adult tissues, this pathway controls vascular permeability, inflammation, and mainly pathological angiogenic responses [410]. Moreover, in association with VEGF, Ang-2 leads to vascular sprouting [409]. Therefore, the deregulation of angiopoietin contributes to the pathogenicity of several diseases, including cancer and AMD.

Faricimab or RG7716

Faricimab, or RG7716, is a related human monoclonal antibody developed by Roche (Basel, Switzerland). This dimeric molecule is composed of an anti-VEGF and an anti-Ang-2 domains. In a mouse model, intravitreal injections of faricimab showed an effectiveness of this dual targeting in CNV [412]. A phase 1 study (NCT01941082, 2013–2016) [413] and phase 2 study (AVENUE, NCT02484690, 2015–2019) [414] have already demonstrated a VA improvement with this mAb. Recently, three phase 3 studies (LUCERNE, NCT03823300; TENAYA, NCT03823287; AVONELLE-X, NCT04777201, 2021) [415,416,417] were initiated to compare combination treatments with aflibercept. Significant gain of VA has been observed for faricimab, similar to aflibercept [418]. Moreover a low rate of intraocular inflammation IOI has been noted without retinal vasculitis or vascular occlusion [415,416,418]. Overall, this compound seems to be particularly promising and could be marketed in the next future.

Nesvacumab or REGN910-3

Nesvacumab, or REGN910-3, is a monoclonal antibody developed by Regeneron Pharmaceuticals (Tarrytown, NY, USA), which specifically targets Ang-2. A phase 1 study (NCT01997164, 2013–2016) [419] has been completed to evaluate its efficacy and safety when administrated in combination with aflibercept against exudative AMD and Diabetic Macular Edema (DME). A phase 2 study (ONYX, NCT02713204, 2016–2019) [420] was performed to compare the therapeutic effects of nesvacumab vs. aflibercept intravitreal injections, but showed inconclusive results.

### 4.5. Miscellaneous Targets

Besides angiogenesis, several other targets have been explored to tackle hallmarks of wet AMD, summarized in Table 10. This includes mainly PPAR and integrin receptors, and anti-immune or anti-inflammatory pathways (e.g., mTOR, TNF-α, complement cascade) [421]. Various small molecules and monoclonal antibodies have been studied for both types of advanced AMD. However, most of these drugs did not go beyond phase 2 clinical trials because of lack of efficacy or tolerance problems for healthy patients, at the exception of anecortave acetate studied in a phase 3 clinical trial in 2012 [422].

### 4.6. Stem Cell Transplant

Human-induced pluripotent stem cells (iPSC)

iPSC was tested in in vivo assays (mice and rats) to evaluate the tumorigenic potential of hiPSC-derived RPE in exudative AMD. This study showed no tumors’ appearance during 6–12 months of monitoring [453]. iPSC was first subretinally transplanted in patients with exudative AMD in 2014 by Riken (Kobe, Japan) [454], to replace or regenerate dead or dying RPE. No side effects were observed, and the vision loss was stabilized [218].

Human embryonic stem cells (hESC)

The implantation of **PF-05206388** (retinal pigment epithelium-derived from human embryonic stem cells) to replace RPE in patients suffering wet AMD has been recently studied through a phase 1 study (NCT01691261, 2012–2019) [455] and a safety follow-up study (NCT03102138, 2017–2020) [456] developed by Moorfields Eye Hospital NHS Foundation Trust. The results have not been posted yet.

Conclusions on treatments for exudative AMD

Exudative AMD is currently one of the major causes of blindness and is globally responsible for 90% of vision loss; this disease is a worldwide serious problem of public health [457]. The anti-VEGF therapies, based on humanized monoclonal antibodies (mAbs) including ranibizumab, bevacizumab, brolucizumab, or dimeric recombinant fusion glycoprotein as aflibercept and combercept, give indisputable benefits for approx. 30% of the patients. However, the lack of predictive factors for the patient’s response, the traumatic effect of repeated intraocular injections, and the long terms side effects, mainly atrophy of the retina [252], render these therapeutic options suboptimal. In line with these considerations, new treatments must emerge in the coming years to offer better care to patients. These future trends may tackle other hallmarks of the disease, such as the inflammatory and immune components as well as the integrin pathways. Other curative strategies, including the use of stem cells, should give a perspective for vision recovery of the vision, and may be a complete cure if used in combination with other treatments. The following table (Table 11) summarizes the current investigated drugs in clinical trials for the treatment of exudative AMD.

## 5. Perspectives

AMD became a real public health concern in the middle of the 20th century due to the increase in life expectancy. This is a complex and poorly understood multi-factorial pathology. Moreover, its two late stages, the dry and wet forms, share the same “AMD” acronym but can be considered as two distinct diseases due to their strong divergences in their own physiopathology (Table 12).

A common feature for the wet and dry AMD forms, remains that all past and current clinically approved treatments are not curative. Due to the “cancer heritage” (the anti-angiogenic therapies in oncology), the only therapeutic option currently available consists in targeting the neovascularization of the disease’s exudative form. This therapy slows down or even stop, in some cases, the invasion of the Bruch’s membrane/RPE complex and the retina by new vessels. In consequence, it reduces the liquid accumulation into the retina and the degeneration of the photoreceptor cells. Therefore, anti-angiogenic treatments are to date still the only treatment, despite limited response rates in patients and possible inconvenience due to the inevitable intravitreal administration.

In general, the emergence of treatments against multi-factorial diseases is a real challenge for clinicians and researchers, due to the complexity of their evolution and the redundant origins of their pathogenicity. In the case of AMD, the challenge for ophthalmologists is greater, as invasive retinal analyses such as biopsies of the different eye segments cannot be performed. This is an obstacle to understanding the mechanisms of ocular diseases and, consequently, to the development of specific and effective therapies against AMD.

To date, the pathogenesis of AMD remains largely unclear. Only risks factors, including genetic background and environmental conditions, have been pointed out. The reasons of its evolution towards its late stages and the causes leading, in some cases, to the switch from the dry to the wet form are still unknown. In addition, the pathogenicity of the neovascularization, which is targeted by the current clinical treatments, remains in some cases questionable. Indeed, during AMD early stages, vessels invasion is restricted to the choroid and neovascularization may contribute to delay the disease progression by providing better oxygen and nutrients supply to the Bruch’s membrane, the RPE and the photoreceptor cells. This supply also balances the stress induced by growing drusens and eventually brings benefits by delaying cell death. Therefore, the precise timing of the anti-angiogenic therapies should be carefully studied to optimize the use of these symptomatic treatments. In summary, a better understanding of AMD pathogenicity associated with an extensive research work is absolutely needed to identify potent future treatments. Those may potentially include drugs directed towards new other targets and/or AMD mechanisms. In particular, the search for an efficient treatment against the dry AMD form explores many options, as illustrated in Table 12. For example, the targeting of drusens appears as an appealing pathway to target as it is a common feature of AMD early stages. Other axes than the regulation of visual cycle, which showed limited efficacy in clinical trials, would be beneficial to investigate. To this end, a precise comprehension of drusens origin and formation may greatly help to develop agents that counteract their accumulation in the retinal and the subretinal spaces. The high levels of pro-inflammatory factors and ROS species seem to be other key factors involved in the pathogenicity of both the atrophic and exudative stages, that could be attenuated by specific therapeutic agents. However, the drugs available until now have only shown limited efficacy in clinical trials. Overall, the clinical treatment of AMD remains essentially based on the use of anti-VEGF drugs, alone or in combination. Gene therapy is also focusing on the expression of anti-VEGF factors. A real challenge remains for the treatment of dry AMD, for which few effective therapeutic options exist. However, the alternative pathway complement seems the most appealing with successful phase 3 clinical trials obtained with APL-2. This validates the relevance of targeting the complement pathway in AMD and the development of new improved drugs interfering with this pathway would be of high interest.

Advanced forms of AMD induce irreversible photoreceptors’ death, underlying the relevance of an early detection of this pathology. Whenever a potent future symptomatic treatment would be found, it may only stop or drastically slow down the disease progression, but it will not lead to the central vision recovery and to a complete cure for patients. Communication and information campaigns directed towards the aged public may lead to early AMD diagnosis, particularly because some tests (Amsler Grid) may be self-performed at home. The exudative AMD form, if untreated, progresses rapidly to blindness and the rapid management of an efficient symptomatic treatment by ophthalmologists is mandatory to keep central vision at an acceptable level for the patient’s daily activities.

Of course, the complete AMD cure implies vision’s recovery, and therefore the total or partial regeneration of atrophied cells. Even though first gene therapy strategies failed to cure AMD, recent progresses in the clinical use of implants containing stem cells may pave the way to a curative regeneration of retina. Several trials dealing with the use of human pluripotent, embryonic or CNS stem cell transplants are currently ongoing for the treatment of atrophic and exudative AMD (Table 12). The first results of their safety and stability, listed in this review, appear very encouraging. However, most of these trials are not fully completed, and long-term outcomes and benefits for patients should be determined in a near future.

In ocular pathologies, another key factor is the drug formulation, which is particularly challenging. Most clinically evaluated drugs are administered by intravitreal injections. Those make possible to reach the posterior fragment of the eye without the need to cross membranes that are quite impermeable to therapeutic agents. In addition, the injection allows precise delivery of the therapeutic agent into a specific eye compartment, and as a result a better control of the administered dose. Therefore, even though these injections are sometimes traumatic for patients, and costly because they can exclusively be performed by an ophthalmologist, the intravitreal administration remains the standard for current ocular anti-neovascularization drugs. Systemic formulations have been less explored for evident concerns of active drug concentration in the eye and minimization of side effects. Yet, it has been used in trials for drugs targeting the visual cycle turn-over (oral capsules for fenretinide and emixustat), or for drugs counteracting β-amyloid formation (intravenous injections of monoclonal antibody). Optimal ocular formulation would be topical administration via eye drops or eye ointments, as those can be by patients used at home, without intervention of a specialist. Nevertheless, it is very challenging to deliver a therapeutic agent into the posterior segment of the eye via this topical administration because contrarily to the hydrophilic ocular fluid of the eye’s anterior segment, the posterior segment is highly hydrophobic. This change may induce severe solubility concerns for the therapeutic agent. Second, the formulation agents’ options and concentrations are limited in the case of eye drops, and many rely on micro emulsions. Moreover, the diffusion of the drug from the anterior to the posterior segment must be quick and efficient since the ocular fluid of the anterior segment is constantly washed and entirely renewed every 30 min. Under these conditions, a large fraction of the therapeutic agent is evacuated by the aqueous humor before entering the posterior segment and makes it difficult to dose the drug in the targeted ocular compartment. However, these challenges have been met with some small molecules, such as the kinase inhibitors sorafenib, pazopanib, and axitinib, which have been tested in trials as eye drops. More research devoted to the drug formulation and delivery to the eye would be very beneficial.

Over the past decade, remarkable efforts have been made in research on AMD by academics and industrialists to better understand its origin and evolution to propose new therapies. Nevertheless, since the validation of the first treatment against its exudative form in 1982 (by laser beam photocoagulation), relatively few drugs have been marketed and all of them are directed toward the retina’s neovascularization. However, drugs targeting the complement pathway appear very promising in the next future after the validation of this mechanism in AMD treatment, as well as combined therapies with anti-VEGFs that would be very relevant to study. In any case, parallel biological research towards a better understanding of the AMD mechanisms is necessary and will undoubtedly contribute to the emergence of new active molecules and biomolecules. Last, drugs’ formulation should be considered to overcome current intravitreal administrations and improve patient’s benefits.

## 6. Conclusions

AMD is a real concern for public health due to the increase in life expectancy. It leads to two distinct diseases, the atrophic (dry) AMD and the exudative (wet) AMD. These two forms are multifactorial, and their physiopathology remain poorly understood. Therefore, to date neither curative nor palliative treatments emerged. Very few therapeutic options exist. In the specific case of the exudative form, anti-VEGF drugs, able to counteract neovascularization, can be proposed to patients. However, this strategy remains very questionable, since it suffers from serious drawbacks including long term side effects and traumatic administration. Overall, the success of the anti-angiogenic treatment is not guaranteed for all treated patients.

In this context strong efforts are made worldwide to reach a better understanding of the AMD pathological processes. The efforts aim also to provide a better understanding of the patient’s response to the different assayed drugs, paving thus the way to a personalized anti-AMD medicine. Last, in this specific case of an ophthalmic disease, drug formulation is of utmost importance for treatment compliance and efficacy (Figure 21).

Due to the recent advances described in this review, one can assume that the emergence of a potential curative treatments for the two AMD forms is no more a chimer but surely requires still strong efforts.

## Data Availability

Not applicable.

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
