# Peer review of "Recent Advances in Age-Related Macular Degeneration Therapies"

_molecules, 2022, doi:10.3390/molecules27165089_

Round 1
Reviewer 1 Report
This is a well-written manuscript. It can be accepted after minor corrections. However, I have the following corrections:-
1. The authors are recommended to add some keywords.
2. The Introduction should not be subdivided.
3. Too many subsections in section 3. It is recommended to reduce it.
4. Carotenoids play vital roles in this context. It is recommended to discuss the role of lutein and other carotenoids clearly.
5. Concluding remarks should be added.
Author Response
- The authors are recommended to add some keywords.
The reviwer is right, and we added some keywords at the beginning of the review.
- The Introduction should not be subdivided.
Thank you to the reviewer for this remark; according to Rewiever’s comments, the subdivision have been removed. - Too many subsections in section 3. It is recommended to reduce it.
According to our point of view, it is not possible to reduce the different sub-sections of the third part of our manuscript. Indeed, each section correspond to a precise therapeutic target, and it seems not suitable to gather theses sub-sections.
- Carotenoids play vital roles in this context. It is recommended to discuss the role of lutein and other carotenoids clearly.
We are grateful to the reviewer for this very good remark. We added in the revised manuscript a reference and two structures highlighted in yellow.
- Concluding remarks should be added.
The reviewer is right, and we introduced a conclusion in the revised manuscript.
Reviewer 2 Report
Thank you authors for this review. I think it is great piece of work discussing Atrophic and Exudative AMD and the current and future treatments (undergoing clinical trials).
Overall I think the work is very comprehensive and easy to read, but I would suggest improving the organisation of the manuscript. Currently it feels like 2 papers are concatinated together. I would like that authors draw more comparisons between two treatments.
I can also see a lot of different drugs/treatments discussed for both pathologies. It would be good to improve the organisation of these. Not sure why authors have chosen to discuss 1 disease and then the other. In my oppinion it would be better to discuss a specific type of treatment for 1 and then other disease and then pull some comonalities and differences for each section. Table comparing both would be also great. I struggled to folow the different drugs and why these speciffic applications are discussed.
Is there any treatments that can be used for both?
Authors did touch a bit on genetics of this disease but I would also like to see if how much the pathologies are driven by genetics and how much by a lifestyle, what can be done in terms of a preventative medicine?
What about the matrikine uses in the treament of AMD? Authors mentioned a few peptides, but I didnt see any specific mentions of matrikines.
Overall great work and I certainly learned new things reading this. Thank you!
Author Response
Reviewer 2
Thank you authors for this review. I think it is great piece of work discussing Atrophic and Exudative AMD and the current and future treatments (undergoing clinical trials).
We are indebted to Rewiever 2 for this kind remark.
Overall, I think the work is very comprehensive and easy to read, but I would suggest improving the organization of the manuscript. Currently it feels like 2 papers are concatenated together. I would like that authors draw more comparisons between two treatments.
Thank you for this comment.
However, making pertinent comparisons between the treatments are quite difficult, as the physiopathologies of the two AMD forms are very different. The dry form is characterized by a cell necrosis mainly induced by drusens formation and accumulation; by a marked contrast, the wet form is characterized by fluid accumulation and by a strong angiogenic context, which is of utmost importance in terms of disease evolution. Last, in both cases, no curative treatment has emerged, yet.
Therefore, targets selected by researchers to slow down de disease process are different, since the expected effects are not the same.
I can also see a lot of different drugs/treatments discussed for both pathologies. It would be good to improve the organization of these.
We are grateful to the Reviewer for this comment.
In the last decades, a plethora of hypothesis emerged, and many therapeutic options have been explored, especially in the case of dry AMD, which is the most common form. In this review, we describe every strategy, which led to a drug assayed in clinical trials. We tried to be as exhaustive as possible, and this induced a lot of section and subsections. Thus, to reach this objective, the section 3 describing the drugs candidates to treat the dry AMD form has been divided into 6 subsections. We assume that this may be a bit confusing for the reader, but this was mandatory to introduce each type of drug candidate.
To counteract the feeling of disorganization of this manuscript, we have introduced in the revised version of our manuscript a table to facilitate the reading of the review.
Not sure why authors have chosen to discuss 1 disease and then the other. In my opinion it would be better to discuss a specific type of treatment for 1 and then other disease and then pull some commonalities and differences for each section. Table comparing both would be also great. I struggled to follow the different drugs and why these specific applications are discussed.
This remark is really interesting.
However, such a reorganization of the manuscript will change in depth the purpose of our review. Indeed, we aim at providing a comprehensive review in which physiopathology is in close contact with the therapeutic options assayed by researchers. In line with this consideration, and due to the strong specificities of each form, it appeared us more suitable to investigate separately and successively the dry and the wet AMD forms.
Nevertheless, to help the reader, we introduced a table in the perspective section, in order to compare the different options.
Are there any treatments that can be used for both?
Thank you to the reviewer for this very pertinent remark.
Inflammation is a common target to treat both dry and wet form of AMD; however, no drugs has been marketed to address inflammation in AMD context, yet. This point has been discussed in the review in section 3.5 describing the different inflammation pathways.
In the case of wet-AMD research is mainly focused on the development of anti-angiogenic drugs. Indeed, researchers have a strong experience in this field du to the advances in the search for such treatment in the case of cancer. This is evidenced in our review since part 4.2 dealing with this specific aspect of the wet-AMD treatment is our longest sub-section.
Authors did touch a bit on genetics of this disease but I would also like to see if how much the pathologies are driven by genetics and how much by a lifestyle, what can be done in terms of a preventative medicine?
Thank you for this remark. We chose to mention genetics and environmental factors without any deeper analysis as none of these factors are considered as treatments.
AMD is etiologically complex, with many epidemiologic and genetic factors that influence susceptibility to risk. Some of these epidemiological risk factors are modifiable, such as body-mass index, smoking cigarettes or blood lipid and cholesterol levels.
Age-related macular degeneration usually does not have a clear-cut pattern of inheritance. Large genome-wide association studies have identified over 30 genes associated with the risk of developing AMD. Variations in two genes have been more closely connected to both developing AMD and whether it progresses to the advanced stages of the disease (mainly atrophic AMD). These are: the complement cascade on chromosome 1; and the ARMS2/HTRA genes on chromosome 10.
Inflammation and immune mechanisms are also part of the pathophysiology of AMD. The connection between complement gene variants (involved in inflammation) and AMD supports this theory. Scientists are intensively researching ARMS2/HTRA1, but still the role of these genes in AMD is not yet understood.
Other genes associated with increased AMD risk are involved in cholesterol and lipid metabolism, collagen production, DNA repair, protein binding, and cell signaling. Research on these genes and their role in disease onset and progression are still underway.
Gene therapy is not available for prevention or management of the disease, so there is no benefit of identifying which genes are involved in any individual’s case of macular degeneration.
Prevention relies on a healthy living with exercise, well-balanced diet and no smoking. Vitamins supplements are also prescribed in early stages of the disease. However, for now, preventive medicine only slightly lower the susceptibility to risk.
What about the matrikine uses in the treament of AMD? Authors mentioned a few peptides, but I didnt see any specific mentions of matrikines.
Thank you to the reviewer for this remark; however, we did not find any strong and fully established link between AMD and Matrikines.